# Extensive-Form Game Solving via Blackwell Approachability on Treeplexes

**Darshan Chakrabarti**
IEOR Department
Columbia University
dc3595@columbia.edu

**Julien Grand-Clément**
ISOM Department
HEC Paris
grand-clement@hec.fr

**Christian Kroer**
IEOR Department
Columbia University
christian.kroer@columbia.edu

## Abstract

We introduce the first algorithmic framework for Blackwell approachability on the sequence-form polytope, the class of convex polytopes capturing the strategies of players in extensive-form games (EFGs). This leads to a new class of regret-minimization algorithms that are stepsize-invariant, in the same sense as the Regret Matching and Regret Matching$^+$ algorithms for the simplex. Our modular framework can be combined with any existing regret minimizer over cones to compute a Nash equilibrium in two-player zero-sum EFGs with perfect recall, through the self-play framework. Leveraging predictive online mirror descent, we introduce *Predictive Treeplex Blackwell$^+$* (PTB$^+$), and show a $O(1/\sqrt{T})$ convergence rate to Nash equilibrium in self-play. We then show how to stabilize PTB$^+$ with a stepsize, resulting in an algorithm with a state-of-the-art $O(1/T)$ convergence rate. We provide an extensive set of experiments to compare our framework with several algorithmic benchmarks, including CFR$^+$ and its predictive variant, and we highlight interesting connections between practical performance and the stepsize-dependence or stepsize-invariance properties of classical algorithms.

## 1 Introduction

In this paper, we focus on solving *Extensive-Form Games* (EFGs). Finding a Nash equilibrium of a two-player zero-sum EFG can be cast as solving

$$\min_{\boldsymbol{x} \in \mathcal{X}} \max_{\boldsymbol{y} \in \mathcal{Y}} \langle \boldsymbol{x}, \boldsymbol{M}\boldsymbol{y} \rangle \tag{1}$$

where the sets $\mathcal{X}, \mathcal{Y}$ are two *sequence-form polytopes* (also referred to as *treeplexes*) representing the strategies $\boldsymbol{x}, \boldsymbol{y}$ of each player, and $\boldsymbol{M}$ is a payoff matrix. EFGs have been successfully used to obtain superhuman performances in several recent poker AI breakthroughs [37, 4, 5]. Many algorithms have been developed based on (1). Since $\mathcal{X}$ and $\mathcal{Y}$ are polytopes, (1) can be formulated as a linear program [38]. However, because $\mathcal{X}$ and $\mathcal{Y}$ themselves have very large dimensions in realistic applications, *first-order methods* (FOMs) and *regret minimization* approaches are preferred for large-scale game solving. FOMs such as the Excessive Gap Technique (EGT, [32]) and Mirror Prox [31] instantiated for EFGs [23, 27] converge to a Nash equilibrium at a rate of $O(1/T)$, where $T$ is the number of iterations. Regret minimization techniques rely on a folk theorem relating the regrets of the players and the duality gap of the average iterates [19]. For instance, predictive online mirror descent with the treeplexes $\mathcal{X}$ and $\mathcal{Y}$ as decision sets achieves a $O(1/T)$ convergence rate [14].

*Counterfactual regret minimization* (CFR) [39] is a regret minimizer for the treeplex that runs regret minimizers *locally*, i.e. directly at the level of the information sets of each player. CFR$^+$, used in virtually all poker AI milestones [37, 30, 5], instantiates the CFR framework with a regret minimizer

38th Conference on Neural Information Processing Systems (NeurIPS 2024).

called *Regret Matching*$^+$ (RM$^+$) [37] and guarantees a $O(1/\sqrt{T})$ convergence rate. The strong empirical performance of CFR$^+$ remains mostly unexplained, since this algorithm does not achieve the fastest theoretical $O(1/T)$ convergence rate. Interestingly, there is a stark contrast between the role of stepsizes in CFR$^+$ versus in other algorithms. CFR$^+$ may use different stepsizes across different infosets, and the iterates of CFR$^+$ do not depend on the values of these stepsizes. We identify this property as *infoset stepsize invariance*. In contrast, the convergence properties of FOMs depend on the choice of a single stepsize used across the entire treeplex, which may be hard to tune in practice.

RM$^+$ is an instantiation of *Blackwell approachability* [3] for the simplex, a versatile framework with connections to online learning [1]. Empirically, using a regret minimizer (over simplexes) based on Blackwell approachability (RM$^+$) is central to the success of CFR$^+$: combining CFR with other local regret minimizers than RM$^+$, e.g., Online Mirror Descent (OMD), leads to much weaker practical performance [6]. This raises the question of whether the performance of CFR$^+$ is mostly explained by the use of Blackwell approachability *on simplexes* (RM$^+$), and if a Blackwell approachability-based algorithm operating *directly on treeplexes*, bypassing the CFR decomposition, could outperform CFR$^+$. Our **goal** in this paper is to address these questions. To do so, we develop the *first* Blackwell approachability-based algorithms for treeplexes, and we provide a new hypothesis for explaining the performance of CFR$^+$. In particular, our **main contributions** are as follows.

**Treeplex Blackwell approachability.** We introduce the first Blackwell approachability-based regret minimizer for treeplexes. Using the self-play framework, we correspondingly get the first framework for solving two-player zero-sum EFGs via Blackwell approachability on treeplexes. Blackwell approachability enables an equivalence between regret minimization over the treeplex $\mathcal{T}$ and over its conic hull cone($\mathcal{T}$), and any existing regret minimizer for cone($\mathcal{T}$) yields a new algorithm for solving EFGs. A crucial advantage of using Blackwell approachability on the treeplex, rather than regret minimization directly on the treeplex, is that it leads to a variety of interesting stepsize properties (e.g. stepsize invariance), which are not achieved by regret minimizers such as OMD on the treeplex.

We then provide several instantiations of our framework. PTB$^+$ (*Predictive Treeplex Blackwell*$^+$, Algorithm 2) combines our framework with predictive OMD over cone($\mathcal{T}$) and achieves a $O(1/\sqrt{T})$ convergence rate. PTB$^+$ is *treeplex stepsize invariant*: its iterates do not change if we rescale all stepsizes by a positive constant. This is a desirable property for practical use, although it is a weaker property than the *infoset* stepsize invariance of CFR$^+$. Smooth PTB$^+$ (Algorithm 3) is a variant of PTB$^+$ ensuring that successive iterates vary smoothly. We show that Smooth PTB$^+$ is the first EFG-solving algorithm based on Blackwell approachability achieving a $O(1/T)$ convergence rate, answering an important open question. Crucially, it is necessary to introduce a stepsize to achieve this faster convergence, and thus Smooth PTB$^+$ is not treeplex stepsize invariant; this is analogous to existing FOM-based $O(1/T)$-methods for solving EFGs. We also consider AdaGradTB$^+$ and AdamTB$^+$, which learn different stepsizes for every dimension of the treeplexes, based on AdaGrad [12] and Adam [25]. We present the convergence properties of our algorithms in Table 1.

**Numerical experiments.** We provide two comprehensive sets of numerical experiments over benchmark EFGs. We find that PTB$^+$ performs the best among all the algorithms introduced in our paper (Figure 4), highlighting the advantage of *treeplex* stepsize invariant algorithms (PTB$^+$) over stepsize-dependent algorithms achieving faster theoretical convergence rate (Smooth PTB$^+$), and over adaptive algorithms learning decreasing stepsizes (AdaGradTB$^+$, AdamTB$^+$). We then compare our best method (PTB$^+$) with CFR$^+$, predictive CFR$^+$ (PCFR$^+$), and predictive OMD (POMD) (Figure 2). We expected PTB$^+$ to perform on par with PCFR$^+$, since PTB$^+$ is stepsize invariant, predictive, and based on Blackwell approachability. However, we find that PCFR$^+$ outperforms all other algorithms. This suggests that *infoset* stepsize invariance is an important property, even more than the *treeplex* stepsize invariance of PTB$^+$. Due to the CFR decomposition, PCFR$^+$ can use different stepsizes at different infosets, where the values of the variables may be of very different magnitudes (typically, smaller for infosets appearing deeper in the treeplex), and PCFR$^+$ does not require tuning these different stepsizes, which may be impossible for large instances. No algorithms appear to consistently outperform the others for the last-iterate performances, and we leave studying this as an open question.

**A new hypothesis on EFG-solving algorithms: the role of stepsize invariance.** Overall, as part of our main contributions, we identify and distinguish the infoset and treeplex stepsize invariance properties, and based on our empirical experiments, we posit that infoset stepsize invariance explains part of the puzzle behind the strong empirical performance of CFR$^+$ and PCFR$^+$. Our results highlight that for practical performance, the stepsize invariance properties may be more important than faster

theoretical convergence rates, which require introducing a stepsize, as for `Smooth PTB`+ or `POMD`. The very strong empirical performance of (predictive) `CFR`+ has been unexplained for a long time and is one of the major open questions in EFG-solving; we view providing a new hypothesis for this phenomenon (infoset stepsize invariance) as important contributions to the EFG-solving community.

| Algorithms | Convergence rate | Stepsize invariance |
|---|---|---|
| CFR+ [37] | $1/\sqrt{T}$ | ✓✓ |
| PCFR+ [16] | $1/\sqrt{T}$ | ✓✓ |
| EGT [26] | $1/T$ | ✗ |
| POMD [14] | $1/T$ | ✗ |
| PTB+ (Algorithm 2) | $1/\sqrt{T}$ | ✓ |
| Smooth PTB+ (Algorithm 3) | $1/T$ | ✗ |
| AdaGradTB+ (Algorithm 6) | $1/\sqrt{T}$ | ✗ |
| AdamTB+ (Algorithm 7) | ? | ✗ |

Table 1: Convergence rates to a Nash equilibrium of a two-player zero-sum EFG for several algorithms. ✓✓ refers to *infoset* stepsize invariance and ✓ refers to *treeplex* stepsize invariance.

## 2 Preliminaries on EFGs

We first provide some background on EFGs and treeplexes.

**Extensive-form games.** Two-player zero-sum extensive-form games (later referred to as *EFGs*) are represented by a game tree and a payoff matrix. Each node of the tree belongs either to one of the players, or to a *chance player*, modeling the random events in the game, e.g., tossing a coin. The players are assigned payoffs at the terminal nodes only. Imperfect information is modeled using *information sets* (*infosets*), which are subsets of nodes of the game tree. A player cannot distinguish between the nodes in a given infoset, and they must take the same action at all these nodes.

**Treeplexes.** The strategy of a player can be described by a polytope called the *treeplex*, also known as the *sequence-form polytope*. The treeplex is constructed as follows. We index the infosets of a player by $\mathcal{J} = \{1, ..., |\mathcal{J}|\}$. The set of actions available at infoset $j \in \mathcal{J}$ is written $\mathcal{A}_j$ with cardinality $|\mathcal{A}_j| = n_j$. We represent choosing action $a \in \mathcal{A}_j$ at infoset $j \in \mathcal{J}$ by a *sequence* $(j, a)$, and we denote by $\mathcal{C}_{ja}$ the set of next infosets reachable from $(j, a)$ (possibly empty if the game terminates). The parent $p_j$ of an infoset $j \in \mathcal{J}$ is the sequence leading to $j$; note that $p_j$ is unique assuming perfect recall. We assume that there is a single root denoted as $\varnothing$ and called the *empty sequence*. If the player does not take any action before reaching $j \in \mathcal{J}$, then by convention $p_j = \varnothing$. Under the perfect recall assumption, the set of infosets has a tree structure: $\mathcal{C}_{ja} \cap \mathcal{C}_{j'a'} = \emptyset$, for all pairs of sequences $(j, a)$ and $(j', a')$ such that $j \neq j', a \neq a'$. This tree is the treeplex and it represents the set of all admissible strategies for a given player. We denote by $n \in \mathbb{N}$ the total number of sequences $(j, a)$ with $j \in \mathcal{J}$ and $a \in \mathcal{A}_j$. With these notations, the treeplex $\mathcal{T}$ of a given player is

$$\mathcal{T} = \{\boldsymbol{x} \in \mathbb{R}_+^{n+1} \mid x_\varnothing = 1, \sum_{a \in \mathcal{A}_j} x_{ja} = x_{p_j}, \forall \, j \in \mathcal{J}\} \tag{2}$$

where the first component $x_\varnothing$ is related to the empty sequence $\varnothing$. A player *makes an observation* to arrive at $j$, if $|\mathcal{C}_{p_j}| > 1$. We define the depth $d$ of a treeplex to be the maximum number of actions and observations that can be made starting at the root until reaching a leaf infoset. Computing a Nash equilibrium of EFGs can be formulated as solving (1) (under the perfect recall assumption), with $\mathcal{X} \subset \mathbb{R}^{n_1+1}$ and $\mathcal{Y} \subset \mathbb{R}^{n_2+1}$ the treeplex of each player, $n_1$ and $n_2$ are the number of sequences of each player, and $\boldsymbol{M} \in \mathbb{R}^{(n_1+1) \times (n_2+1)}$ the payoff matrix such that for a pair of strategy $(\boldsymbol{x}, \boldsymbol{y}) \in \mathcal{X} \times \mathcal{Y}$, $\langle \boldsymbol{x}, \boldsymbol{M}\boldsymbol{y} \rangle$ is the expected value that the second player receives from the first player.

**Regret minimization and self-play framework.** A *regret minimizer* `Regmin` over a decision set $\mathcal{Z} \subset \mathbb{R}^d$ is an algorithm such that, at every iteration, `Regmin` chooses a decision $\boldsymbol{z}^t \in \mathcal{Z}$, a *loss vector* $\boldsymbol{\ell} \in \mathbb{R}^d$ is observed, and the scalar loss $\langle \boldsymbol{\ell}^t, \boldsymbol{x}^t \rangle$ is incurred. A regret minimizer ensures that the *regret* $\mathsf{Reg}^T = \max_{\hat{\boldsymbol{z}} \in \mathcal{Z}} \sum_{t=1}^{T} \langle \boldsymbol{\ell}^t, \boldsymbol{z}^t - \hat{\boldsymbol{z}} \rangle$ grows at most as $O(\sqrt{T})$. As an example, *predictive online mirror descent* (`POMD`, [34]) generates a sequence of decisions $\boldsymbol{z}_1, ..., \boldsymbol{z}_T \in \mathcal{Z}$ as follows:

$$\boldsymbol{z}_t = \Pi_{\mathcal{Z}}(\hat{\boldsymbol{z}}_t - \eta \boldsymbol{m}_t), \hat{\boldsymbol{z}}_{t+1} = \Pi_{\mathcal{Z}}(\hat{\boldsymbol{z}}_t - \eta \boldsymbol{\ell}_t) \tag{3}$$

with $\boldsymbol{m}_1, ..., \boldsymbol{m}_T \in \mathbb{R}^d$ some predictions of the losses $\boldsymbol{\ell}_1, ..., \boldsymbol{\ell}_T \in \mathbb{R}^d$, and where we write the orthogonal projection of $\boldsymbol{y} \in \mathbb{R}^d$ onto $\mathcal{Z}$ as $\Pi_{\mathcal{Z}}(\boldsymbol{y}) := \arg\min_{\boldsymbol{z} \in \mathcal{Z}} \|\boldsymbol{z} - \boldsymbol{y}\|_2$.

The *self-play framework* solves EFGs via regret minimization. The players compute two sequences of strategies $\boldsymbol{x}_1, ..., \boldsymbol{x}_T$ and $\boldsymbol{y}_1, ..., \boldsymbol{y}_T$ such that, at iteration $t \geq 1$, the first player observes its loss vector $\boldsymbol{M}\boldsymbol{y}_{t-1}$ and the second player observes its loss vector $-\boldsymbol{M}^\top \boldsymbol{x}_{t-1}$. Each player computes their current strategies $\boldsymbol{x}_t \in \mathcal{X}$ and $\boldsymbol{y}_t \in \mathcal{Y}$ via regret minimization. A well-known theorem states that the duality gap of the average of the iterates is bounded by the sum of the average regrets of the players.

**Proposition 2.1** ([19]). *Let $\boldsymbol{x}_1, ..., \boldsymbol{x}_T \in \mathcal{X}$ and $\boldsymbol{y}_1, ..., \boldsymbol{y}_T \in \mathcal{Y}$ be computed in the self-play framework. Let $(\bar{\boldsymbol{x}}_T, \bar{\boldsymbol{y}}_T) = \frac{1}{T}\sum_{t=1}^{T}(\boldsymbol{x}_t, \boldsymbol{y}_t)$. Then, for $\mathsf{Reg}_1^T$ and $\mathsf{Reg}_2^T$ the regret of each player,*

$$\max_{\hat{\boldsymbol{y}} \in \mathcal{Y}} \langle \bar{\boldsymbol{x}}_T, \boldsymbol{M}\hat{\boldsymbol{y}} \rangle - \min_{\hat{\boldsymbol{x}} \in \mathcal{X}} \langle \hat{\boldsymbol{x}}, \boldsymbol{M}\bar{\boldsymbol{y}}_T \rangle = \left( \mathsf{Reg}_1^T + \mathsf{Reg}_2^T \right)/T.$$

We present more details on the self-play framework in Appendix A.

**CFR and Regret Matching$^+$.** *Counterfactual Regret minimization* (CFR, [39]) runs independent regret minimizers with counterfactual losses at each infoset of the treeplexes. This considerably simplifies the optimization problem, since the decision set at each infoset $j \in \mathcal{J}$ is the simplex over the set of next available actions $\Delta^{n_j} := \{\boldsymbol{x} \in \mathbb{R}_+^{n_j} \mid \sum_{i=1}^{n_j} x_i = 1\}$. In the CFR framework, the regret of each player (over the treeplex) is bounded by the maximum of the local regrets incurred at each infoset. Therefore, CFR combined with any regret minimizer over the simplex converges to a Nash equilibrium at a rate of $O(1/\sqrt{T})$. We refer to Appendix B for more details. Combining CFR with a local regret minimizer called *Regret Matching$^+$* (RM$^+$, [37]) along with alternation and linear averaging yields an algorithm called CFR$^+$, which has been observed to attain strong practical performance compared to theoretically-faster methods [27]. Crucially, RM$^+$ can only be implemented on the simplex and not for other decision sets, and proceeds as follows: given a sequence of loss $\boldsymbol{\ell}_1, ..., \boldsymbol{\ell}_T \in \mathbb{R}^d$, RM$^+$ maintains a sequence $\boldsymbol{R}_1, ..., \boldsymbol{R}_T \in \mathbb{R}^d$ such that $\boldsymbol{R}_1 = \boldsymbol{0}$ and

$$\boldsymbol{x}_t = \boldsymbol{R}_t/\|\boldsymbol{R}_t\|_1, \quad \boldsymbol{R}_{t+1} = \Pi_{\mathbb{R}_+^d}(\boldsymbol{R}_t - \eta\boldsymbol{g}(\boldsymbol{x}_t, \boldsymbol{\ell}_t)) \tag{4}$$

with $\eta > 0$ and $\boldsymbol{0}/0 := (1/d)\boldsymbol{1}$ for $\boldsymbol{1} := (1, ..., 1) \in \mathbb{R}^d$, and, for $\boldsymbol{x}, \boldsymbol{\ell} \in \mathbb{R}^d$,

$$\boldsymbol{g}(\boldsymbol{x}, \boldsymbol{\ell}) := \boldsymbol{\ell} - \langle \boldsymbol{x}, \boldsymbol{\ell} \rangle \boldsymbol{1}. \tag{5}$$

RM$^+$ is *stepsize invariant*: $\boldsymbol{x}_1, ..., \boldsymbol{x}_T$ are independent of $\eta$, since $\boldsymbol{x}_t = \boldsymbol{R}_t/\|\boldsymbol{R}_t\|_1$ and $\eta$ only rescales the entire sequence $\boldsymbol{R}_1, ..., \boldsymbol{R}_T$. Since CFR$^+$ runs RM$^+$ at each infoset independently, CFR$^+$ is *infoset stepsize invariant*: there may be different stepsizes across different infosets and the iterates of CFR$^+$ do not depend on them, which is desirable for large-scale EFGs where stepsize tuning may be difficult.

RM$^+$ can be interpreted as an instantiation of *Blackwell approachability* [3, 1], where the goal of the decision maker is to compute the sequence of strategies $\boldsymbol{x}_1, ..., \boldsymbol{x}_T \in \Delta^d$ to ensure that the auxiliary sequence $\boldsymbol{R}_T/T \in \mathbb{R}_+^d$ approaches the *target set* $\mathbb{R}_-^d$ as $T \to +\infty$. Since $\boldsymbol{R}_t \in \mathbb{R}_+^d$, this is equivalent to ensuring that $\lim_{T\to+\infty} \boldsymbol{R}_T/T = \boldsymbol{0}$. The vector $\boldsymbol{g}(\boldsymbol{x}, \boldsymbol{\ell})$ is interpreted as an instantaneous loss for the approachability instance. As an instantiation of Blackwell approachability, at each iteration RM$^+$ computes an orthogonal projection onto the *conic hull* of the decision set:

$$\mathbb{R}_+^d = \mathsf{cone}(\Delta^d) \tag{6}$$

with $\mathsf{cone}(\mathcal{Z}) := \{\alpha\boldsymbol{x} \mid \boldsymbol{x} \in \mathcal{Z}, \alpha \geq 0\}$ for a set $\mathcal{Z}$. The function $\boldsymbol{R} \mapsto \boldsymbol{R}/\|\boldsymbol{R}\|_1$ is based on

$$\Delta^d \subset \{\boldsymbol{x} \in \mathbb{R}^d \mid \langle \boldsymbol{x}, \boldsymbol{1} \rangle = 1\}. \tag{7}$$

Since for $\boldsymbol{R} \in \mathbb{R}_+^d$, $\langle \boldsymbol{R}, \boldsymbol{1} \rangle = \|\boldsymbol{R}\|_1$, then $\boldsymbol{x}_t = \boldsymbol{R}_t/\|\boldsymbol{R}_t\|_1$ can be written $\boldsymbol{x}_t = \boldsymbol{R}_t/\langle \boldsymbol{R}_t, \boldsymbol{1} \rangle$, with $\boldsymbol{1}$ a vector such that the decision set $\Delta^d$ satisfies (7). This ensures that

$$\langle \boldsymbol{R}_t, \boldsymbol{g}(\boldsymbol{x}_t, \boldsymbol{\ell}) \rangle = 0, \forall\, \boldsymbol{\ell} \in \mathbb{R}^d. \tag{8}$$

We provide an illustration of the dynamics of RM$^+$ in Figure 1. Equation (8) is known as a *hyperplane forcing condition* and is a key ingredient in any Blackwell approachability-based algorithm; it ensures that the vector $\boldsymbol{R}_T$ grows at most at a rate of $O(\sqrt{T})$ so that $\lim_{T\to+\infty} \boldsymbol{R}_T/T = \boldsymbol{0}$. We refer to [33, 22] and to Appendix C for more details on Blackwell approachability.

# 3  Blackwell Approachability on Treeplexes

In this section we introduce a modular regret minimization framework for the treeplex based on Blackwell approachability. This framework can be used as a regret minimizer over $\mathcal{T}$ in the self-play framework (described in the previous section and in Appendix A) to obtain an algorithm for solving EFGs. Our algorithms are based on the fact that for $\mathcal{T} \subset \mathbb{R}^{n+1}$ a treeplex as defined in (2), we have

$$\mathcal{T} \subset \{ \boldsymbol{x} \in \mathbb{R}^{n+1} \mid \langle \boldsymbol{x}, \boldsymbol{a} \rangle = 1 \} \tag{9}$$

for $\boldsymbol{a} = (1, \boldsymbol{0}) \in \mathbb{R}^{n+1}$ with $\boldsymbol{0} = (0, ..., 0) \in \mathbb{R}^n$. This property is analogous to (7) for the simplex. With this analogy in mind, we define $\mathcal{C} \subset \mathbb{R}^{n+1}$ and $\boldsymbol{f}(\boldsymbol{x}, \boldsymbol{\ell}) \in \mathbb{R}^{n+1}$ as, for $\boldsymbol{x}, \boldsymbol{\ell} \in \mathbb{R}^{n+1}$,

$$\mathcal{C} := \mathsf{cone}(\mathcal{T}) \tag{10}$$
$$\boldsymbol{f}(\boldsymbol{x}, \boldsymbol{\ell}) := \boldsymbol{\ell} - \langle \boldsymbol{x}, \boldsymbol{\ell} \rangle \boldsymbol{a}. \tag{11}$$

Equation (10) and Equation (11) are analogous to (6) and (5). The cone $\mathcal{C}$ and the vector $\boldsymbol{f}(\boldsymbol{x}, \boldsymbol{\ell})$ play a similar role for $\mathcal{T}$ as $\mathbb{R}_+^d$ and $\boldsymbol{g}(\boldsymbol{x}, \boldsymbol{\ell})$ play for $\Delta^d$ in RM⁺. Our framework is described in Algorithm 1 and relies on running a regret minimizer Regmin over $\mathcal{C} = \mathsf{cone}(\mathcal{T})$ against the losses $\boldsymbol{f}(\boldsymbol{x}_t, \boldsymbol{\ell}_t)$ to obtain a regret minimizer over $\mathcal{T}$ against the losses $\boldsymbol{\ell}_t$, for $t \geq 1$.

---

**Algorithm 1** Blackwell approachability on the treeplex

1: **Input**: A regret minimizer Regmin with decision set $\mathcal{C}$
2: **Initialization**: $\boldsymbol{R}_1 = \boldsymbol{0} \in \mathbb{R}^{n+1}$
3: **for** $t = 1, \ldots, T$ **do**
4:      $\boldsymbol{x}_t = \boldsymbol{R}_t / \langle \boldsymbol{R}_t, \boldsymbol{a} \rangle$
5:      Observe the loss vector $\boldsymbol{\ell}_t \in \mathbb{R}^{n+1}$
6:      Regmin observes $\boldsymbol{f}(\boldsymbol{x}_t, \boldsymbol{\ell}_t) \in \mathbb{R}^{n+1}$
7:      $\boldsymbol{R}_{t+1} = \mathsf{Regmin}\,(\cdot)$

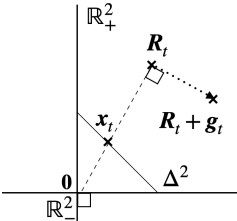

Figure 1: RM⁺ in $\mathbb{R}_+^2$, with $\boldsymbol{g}_t = \boldsymbol{g}(\boldsymbol{x}_t, \boldsymbol{\ell}_t)$.

---

By convention that $\boldsymbol{0}/0$ is the uniform strategy for the treeplex. Algorithm 1 is the first Blackwell approachability-based algorithm operating on the entire treeplex (in contrast to CFR⁺ which relies on Blackwell approachability locally at the infosets level). We first describe some important properties of Algorithm 1:

*Feasibility of the iterates.* Algorithm 1 produces feasible strategies, i.e., $\boldsymbol{x}_t \in \mathcal{T}, \forall\, t \geq 1$. Indeed, since Regmin is a regret minimizer with $\mathcal{C}$ as the decision set, $\boldsymbol{R}_t \in \mathsf{cone}(\mathcal{T})$, i.e., $\boldsymbol{R}_t = \alpha \boldsymbol{z}$ with $\alpha \in \mathbb{R}_+$ and $\boldsymbol{z} \in \mathcal{T}$. From (9), we have $\langle \boldsymbol{z}, \boldsymbol{a} \rangle = 1$. Therefore, $\boldsymbol{x}_t = \frac{\boldsymbol{R}_t}{\langle \boldsymbol{R}_t, \boldsymbol{a} \rangle} = \frac{\alpha \boldsymbol{z}}{\alpha \langle \boldsymbol{z}, \boldsymbol{a} \rangle} = \boldsymbol{z} \in \mathcal{T}$. This is analogous to RM⁺, where $\boldsymbol{x}_t$ is proportional to $\boldsymbol{R}_t$, see (4) and Figure 1.

*Hyperplane forcing.* For any $t \in \mathbb{N}$ we have

$$\langle \boldsymbol{R}_t, f(\boldsymbol{x}_t, \boldsymbol{\ell}) \rangle = 0, \forall\, \boldsymbol{\ell} \in \mathbb{R}^{n+1}. \tag{12}$$

The hyperplane forcing equation (12) is a crucial component of algorithms based on Blackwell approachability. It ensures that $\|\boldsymbol{R}_t\|_2 = O(\sqrt{T})$. Equation (12) is analogous to (8) for RM⁺ and follows from $\boldsymbol{x}_t = \frac{\boldsymbol{R}_t}{\langle \boldsymbol{R}_t, \boldsymbol{a} \rangle}$, so that

$$\langle \boldsymbol{R}_t, \boldsymbol{f}(\boldsymbol{x}_t, \boldsymbol{\ell}) \rangle = \langle \boldsymbol{R}_t, \boldsymbol{\ell} \rangle - \langle \boldsymbol{x}_t, \boldsymbol{\ell} \rangle \langle \boldsymbol{R}_t, \boldsymbol{a} \rangle = \langle \boldsymbol{R}_t, \boldsymbol{\ell} \rangle - \langle \frac{\boldsymbol{R}_t}{\langle \boldsymbol{R}_t, \boldsymbol{a} \rangle}, \boldsymbol{\ell} \rangle \langle \boldsymbol{R}_t, \boldsymbol{a} \rangle = \langle \boldsymbol{R}_t, \boldsymbol{\ell} \rangle - \langle \boldsymbol{R}_t, \boldsymbol{\ell} \rangle = 0.$$

*Regret minimization over $\mathcal{T}$.* Algorithm 1 always yields a regret minimizer over the treeplex $\mathcal{T}$, i.e., it ensures that the regret of $\boldsymbol{x}_1, ..., \boldsymbol{x}_T \in \mathcal{T}$ against any $\boldsymbol{\ell}_1, ..., \boldsymbol{\ell}_T \in \mathbb{R}^{n+1}$ is bounded by $O(\sqrt{T})$. The proof is instructive and shows a central component to Blackwell approachability-based algorithms: minimizing regret over $\mathcal{T}$ can be achieved by minimizing regret over $\mathsf{cone}(\mathcal{T})$.

**Proposition 3.1.** *Let* Regmin *be a regret minimizer with $\mathcal{C}$ as the decision set. Let $\boldsymbol{x}_1, ..., \boldsymbol{x}_T \in \mathcal{T}$ be computed by Algorithm 1. Then* $\max_{\hat{\boldsymbol{x}} \in \mathcal{T}} \sum_{t=1}^T \langle \boldsymbol{x}_t - \hat{\boldsymbol{x}}, \boldsymbol{\ell}_t \rangle = O(\sqrt{T})$.

*Proof.* Let $\hat{\boldsymbol{x}} \in \mathcal{T}$ and let us write $\hat{\boldsymbol{R}} = \hat{\boldsymbol{x}}$. We have

$$\sum_{t=1}^{T} \langle \boldsymbol{x}_t - \hat{\boldsymbol{x}}, \boldsymbol{\ell}_t \rangle = \sum_{t=1}^{T} \langle -\hat{\boldsymbol{x}}, \boldsymbol{f}(\boldsymbol{x}_t, \boldsymbol{\ell}_t) \rangle = \sum_{t=1}^{T} \langle -\hat{\boldsymbol{R}}, \boldsymbol{f}(\boldsymbol{x}_t, \boldsymbol{\ell}_t) \rangle = \sum_{t=1}^{T} \langle \boldsymbol{R}_t - \hat{\boldsymbol{R}}, \boldsymbol{f}(\boldsymbol{x}_t, \boldsymbol{\ell}_t) \rangle$$

where the first equality follows from the definition of $\boldsymbol{f}(\boldsymbol{x}_t, \boldsymbol{\ell}_t)$ and $\langle \boldsymbol{z}, \boldsymbol{a} \rangle = 1$ for any $\boldsymbol{z} \in \mathcal{T}$, the second equality is because $\hat{\boldsymbol{x}} = \hat{\boldsymbol{R}}$, and the last equality follows from the hyperplane forcing condition (12). Now note that $\sum_{t=1}^{T} \langle \boldsymbol{R}_t - \hat{\boldsymbol{R}}, \boldsymbol{f}(\boldsymbol{x}_t, \boldsymbol{\ell}_t) \rangle$ is the regret of a regret minimizer Regmin choosing $\boldsymbol{R}_1, ..., \boldsymbol{R}_T$ in the decision set $\mathcal{C} := \mathrm{cone}(\mathcal{T})$ against a sequence of loss $\boldsymbol{f}(\boldsymbol{x}_1, \boldsymbol{\ell}_1), ..., \boldsymbol{f}(\boldsymbol{x}_T, \boldsymbol{\ell}_T)$ and a comparator $\hat{\boldsymbol{R}} \in \mathrm{cone}(\mathcal{T})$. Therefore, $\sum_{t=1}^{T} \langle \boldsymbol{R}_t - \hat{\boldsymbol{R}}, \boldsymbol{f}(\boldsymbol{x}_t, \boldsymbol{\ell}_t) \rangle = O(\sqrt{T})$. $\square$

**Remark 3.2.** *In their seminal paper, Abernethy et al. [1] show a general reduction from regret minimization to Blackwell approachability for compact convex decision sets. Our reduction from Algorithm 1 builds upon the ideas in [1], but our reduction is different and exploits the structure of treeplexes. Additionally, [1] focuses on the case of adversarial losss, whereas we focus on solving EFGs, where stepsize invariance properties is crucial and where we can prove fast $O(1/T)$ convergence rates. We provide a more detailed comparison with [1] in Appendix C.*

# 4   Instantiations of Algorithm 1

We can instantiate Algorithm 1 with any regret minimizer over $\mathcal{C}$ to obtain various properties such as stepsize invariance or achieving $O(1/T)$ convergence rate. We show next how to do so.

**Predictive Treeplex Blackwell$^+$ (PTB$^+$).** We first introduce *Predictive Treeplex Blackwell$^+$* (Algorithm 2), combining Algorithm 1 with POMD with $\mathcal{C}$ as a decision set.

| **Algorithm 2** PTB$^+$ | **Algorithm 3** Smooth PTB$^+$ |
|---|---|
| 1: **Input**: $\eta > 0$, $\boldsymbol{m}_1, ..., \boldsymbol{m}_T \in \mathbb{R}^{n+1}$ | 1: **Input**: $\eta > 0$, $\boldsymbol{m}_1, ..., \boldsymbol{m}_T \in \mathbb{R}^{n+1}$ |
| 2: **Initialization**: $\hat{\boldsymbol{R}}_1 = \boldsymbol{0} \in \mathbb{R}^{n+1}$ | 2: **Initialization**: $\hat{\boldsymbol{R}}_1 = \boldsymbol{0} \in \mathbb{R}^{n+1}$ |
| 3: **for** $t = 1, \ldots, T$ **do** | 3: **for** $t = 1, \ldots, T$ **do** |
| 4:     $\boldsymbol{R}_t \in \Pi_{\mathcal{C}}\left(\hat{\boldsymbol{R}}_t - \eta \boldsymbol{m}_t\right)$ | 4:     $\boldsymbol{R}_t \in \Pi_{\mathcal{C}_{\geq}}\left(\hat{\boldsymbol{R}}_t - \eta \boldsymbol{m}_t\right)$ |
| 5:     $\boldsymbol{x}_t = \boldsymbol{R}_t / \langle \boldsymbol{R}_t, \boldsymbol{a} \rangle$ | 5:     $\boldsymbol{x}_t = \boldsymbol{R}_t / \langle \boldsymbol{R}_t, \boldsymbol{a} \rangle$ |
| 6:     Observe the loss vector $\boldsymbol{\ell}_t \in \mathbb{R}^{n+1}$ | 6:     Observe the loss vector $\boldsymbol{\ell}_t \in \mathbb{R}^{n+1}$ |
| 7:     $\hat{\boldsymbol{R}}_{t+1} \in \Pi_{\mathcal{C}}\left(\hat{\boldsymbol{R}}_t - \eta \boldsymbol{f}(\boldsymbol{x}_t, \boldsymbol{\ell}_t)\right)$ | 7:     $\hat{\boldsymbol{R}}_{t+1} \in \Pi_{\mathcal{C}_{\geq}}\left(\hat{\boldsymbol{R}}_t - \eta \boldsymbol{f}(\boldsymbol{x}_t, \boldsymbol{\ell}_t)\right)$ |

We start by highlighting a crucial property of PTB$^+$, *treeplex stepsize invariance.* The sequence of iterates $\boldsymbol{x}_1, ..., \boldsymbol{x}_T$ generated by Algorithm 2 is independent of the choice of the stepsize $\eta > 0$, that only rescales the sequences $\hat{\boldsymbol{R}}_1, ..., \hat{\boldsymbol{R}}_T$ and $\boldsymbol{R}_1, ..., \boldsymbol{R}_T$, the orthogonal projection onto a cone is positively homogeneous of degree 1: $\Pi_{\mathcal{C}}(\eta \boldsymbol{z}) = \eta \Pi_{\mathcal{C}}(\boldsymbol{z})$ for $\eta > 0$ and $\boldsymbol{z} \in \mathbb{R}^{n+1}$, and the function $\boldsymbol{R} \mapsto \boldsymbol{R} / \langle \boldsymbol{R}, \boldsymbol{a} \rangle$ is scale-invariant: $\frac{(\eta \boldsymbol{R})}{\langle (\eta \boldsymbol{R}), \boldsymbol{a} \rangle} = \frac{\boldsymbol{R}}{\langle \boldsymbol{R}, \boldsymbol{a} \rangle}$ for $\eta > 0$ and $\boldsymbol{R} \in \mathbb{R}^{n+1}$. We provide a rigorous statement in the following proposition and we present the proof in Appendix D.

**Proposition 4.1.** *The sequence $\boldsymbol{x}_1, ..., \boldsymbol{x}_T$ computed by PTB$^+$ is independent on the stepsize $\eta > 0$.*

Treeplex stepsize invariance is a crucial property, since in large EFGs, stepsize tuning is difficult and resource-consuming. This is the main advantage of using Blackwell approachability: running POMD directly on the treeplex $\mathcal{T}$ does not result in a stepsize invariant algorithm, whereas PTB$^+$ runs POMD on $\mathrm{cone}(\mathcal{T})$ and is stepsize invariant. To our knowledge, CFR$^+$ and PCFR$^+$ are the only other treeplex stepsize invariant algorithms for solving EFGs. In fact, they satisfy a stronger *infoset stepsize invariance* property: different stepsizes can be used at different infosets, and the iterates do not depend on their values. We discuss the relation between PTB$^+$ and known instantiations of Blackwell approachability over the simplex (RM$^+$ and CBA$^+$ [22]) in Appendix E.

From Proposition 3.1 and the regret bounds on POMD (see for instance section 3.1.1 in [34] or section 6 in [16]), we obtain the following proposition. We define $\Omega \in \mathbb{R}_+$ as $\Omega := \max_{\boldsymbol{x} \in \mathcal{T}} \|\boldsymbol{x}\|_2$.

**Proposition 4.2.** *Let $\boldsymbol{x}_1, ..., \boldsymbol{x}_T$ be computed by* PTB$^+$. *Then* $\max_{\hat{\boldsymbol{x}} \in \mathcal{T}} \sum_{t=1}^{T} \langle \boldsymbol{x}_t - \hat{\boldsymbol{x}}, \boldsymbol{\ell}_t \rangle \leq \Omega \sqrt{\sum_{t=1}^{T} \|\boldsymbol{f}(\boldsymbol{x}_t, \boldsymbol{\ell}_t) - \boldsymbol{m}_t\|_2^2}$.

From Proposition 4.2, PTB$^+$ is a regret minimizer over treeplexes, and we can combine it with the self-play framework to solve EFGs, as shown in the next corollary. We use the notations $d := \max\{n, m\} + 1, \hat{\Omega} := \max\{\|\boldsymbol{z}\|_2 \mid \boldsymbol{z} \in \mathcal{X} \cup \mathcal{Y}\}, \|\boldsymbol{M}\|_2 := \sup_{\boldsymbol{v} \neq \boldsymbol{0}} \frac{\|\boldsymbol{M}\boldsymbol{v}\|_2}{\|\boldsymbol{v}\|_2}$.

**Corollary 4.3.** *Let $(\boldsymbol{x}_t)_{t\geq 1}$ and $(\boldsymbol{y}_t)_{t\geq 1}$ be the sequence of strategies computed by both players employing* PTB$^+$ *in the self-play framework, with previous losses as predictions:* $\boldsymbol{m}_t^x = \boldsymbol{f}(\boldsymbol{x}_{t-1}, \boldsymbol{M}\boldsymbol{y}_{t-1}), \boldsymbol{m}_t^y = \boldsymbol{f}(\boldsymbol{y}_{t-1}, -\boldsymbol{M}^\top \boldsymbol{x}_{t-1})$. *Let $(\bar{\boldsymbol{x}}_T, \bar{\boldsymbol{y}}_T) = \frac{1}{T} \sum_{t=1}^{T} (\boldsymbol{x}_t, \boldsymbol{y}_t)$. Then*

$$\max_{\boldsymbol{y} \in \mathcal{Y}} \langle \bar{\boldsymbol{x}}_T, \boldsymbol{M}\boldsymbol{y} \rangle - \min_{\boldsymbol{x} \in \mathcal{X}} \langle \boldsymbol{x}, \boldsymbol{M}\bar{\boldsymbol{y}}_T \rangle \leq \frac{\hat{\Omega}^3 \sqrt{d} \sqrt{\|\boldsymbol{M}\|_2}}{\sqrt{T}}.$$

Finally, we can efficiently compute the orthogonal projection onto $\mathcal{C}$, since $\mathcal{C}$ admits the following simple formulation of as a polytope: $\mathcal{C} = \{\boldsymbol{x} \in \mathbb{R}_+^{n+1} \mid \sum_{a \in \mathcal{A}_j} x_{ja} = x_{p_j}, \forall j \in \mathcal{J}\}$.

**Proposition 4.4.** *Let $\mathcal{T}$ be a treeplex with depth $d$, number of sequences $n$, number of leaf sequences $l$, and number of infosets $m$. The orthogonal projection $\Pi_{\mathcal{C}}(\boldsymbol{y})$ of a point $\boldsymbol{y} \in \mathbb{R}^{n+1}$ onto $\mathcal{C} = \mathrm{cone}(\mathcal{T})$ can be computed in $O(dn \log(l + m))$ arithmetic operations.*

**A stable algorithm:** Smooth PTB$^+$. We now modify PTB$^+$ to obtain faster convergence rates. The $O(1/\sqrt{T})$ average convergence rate of PTB$^+$ may seem surprising since in the *matrix game* setting, POMD over the simplexes obtains a $O(1/T)$ average convergence [36]. This discrepancy comes from PTB$^+$ running POMD *on the set* $\mathcal{C} = \mathrm{cone}(\mathcal{T})$ instead of the original decision set $\mathcal{T}$, so that the Lipschtiz continuity of the loss function and the classical *RVU bounds* (Regret Bounded by Variation in Utilities, see Equation (1) in [36]), central to proving the fast convergence of predictive algorithms, may not hold. For PTB$^+$, the Lipschitz continuity of the loss $\boldsymbol{R} \mapsto \boldsymbol{f}(\boldsymbol{x}, \boldsymbol{\ell})$ with $\boldsymbol{x} = \boldsymbol{R}/\langle \boldsymbol{R}, \boldsymbol{a} \rangle$ depends on the Lipschitz continuity of the decision function $\boldsymbol{R} \mapsto \boldsymbol{R}/\langle \boldsymbol{R}, \boldsymbol{a} \rangle$ over $\mathcal{C}$, which we analyze next.

**Proposition 4.5.** *Let $\boldsymbol{R}_1, \boldsymbol{R}_2 \in \mathrm{cone}(\mathcal{T})$. Then $\left\| \frac{\boldsymbol{R}_1}{\langle \boldsymbol{R}_1, \boldsymbol{a} \rangle} - \frac{\boldsymbol{R}_2}{\langle \boldsymbol{R}_2, \boldsymbol{a} \rangle} \right\|_2 \leq \frac{\Omega \cdot \|\boldsymbol{R}_1 - \boldsymbol{R}_2\|_2}{\max\{\langle \boldsymbol{R}_1, \boldsymbol{a} \rangle, \langle \boldsymbol{R}_2, \boldsymbol{a} \rangle\}}$.*

We present the proof of Proposition 4.5 in Appendix F. Proposition 4.5 shows that when the vector $\boldsymbol{R}$ is such that $\langle \boldsymbol{R}, \boldsymbol{a} \rangle$ is small, the decision function $\boldsymbol{R} \mapsto \boldsymbol{R}/\langle \boldsymbol{R}, \boldsymbol{a} \rangle$ may vary rapidly, an issue known as *instability* and also observed for a predictive variant of RM$^+$ [18]. To ensure the Lipschitzness of the decision function, we can ensure that $\boldsymbol{R}_t$ and $\hat{\boldsymbol{R}}_t$ always belong to the *stable region* $\mathcal{C}_{\geq}$:

$$\mathcal{C}_{\geq} := \mathrm{cone}(\mathcal{T}) \cap \{\boldsymbol{R} \in \mathbb{R}^{n+1} \mid \langle \boldsymbol{R}, \boldsymbol{a} \rangle \geq R_0\}$$

for $R_0 > 0$, and we recover Lipschitz continuity over $\mathcal{C}_{\geq}$:

$$\left\| \frac{\boldsymbol{R}_1}{\langle \boldsymbol{R}_1, \boldsymbol{a} \rangle} - \frac{\boldsymbol{R}_2}{\langle \boldsymbol{R}_2, \boldsymbol{a} \rangle} \right\|_2 \leq \frac{\Omega}{R_0} \|\boldsymbol{R}_1 - \boldsymbol{R}_2\|_2, \forall \boldsymbol{R}_1, \boldsymbol{R}_2 \in \mathcal{C}_{\geq}.$$

This leads us to introduce Smooth PTB$^+$(Algorithm 3), a variant of PTB$^+$,where $\boldsymbol{R}_t$ and $\hat{\boldsymbol{R}}_t$ always belong to $\mathcal{C}_{\geq}$. For Smooth PTB$^+$,$\boldsymbol{x}_t \in \mathcal{T}$ since $\boldsymbol{R}_t \in \mathcal{C}_{\geq} \subset \mathrm{cone}(\mathcal{T})$, and we also have the hyperplane forcing property (12), which only depends on $\boldsymbol{x}_t = \boldsymbol{R}_t/\langle \boldsymbol{R}_t, \boldsymbol{a} \rangle$. However, Smooth PTB$^+$ is not treeplex stepsize invariant, because the orthogonal projections are onto $\mathcal{C}_{\geq}$, which is not a cone. Note that $\mathcal{C}_{\geq}$ admits a simple polytope formulation:

$$\mathcal{C}_{\geq} = \{\boldsymbol{x} \in \mathbb{R}_+^{n+1} \mid x_{\varnothing} \geq R_0, \sum_{a \in \mathcal{A}_j} x_{ja} = x_{p_j}, \forall j \in \mathcal{J}\}$$

so the complexity of computing the orthogonal projection onto $\mathcal{C}_{\geq}$ is the same as computing the orthogonal projection onto $\mathcal{C}$. We provide a proof in Appendix H. We now show that Smooth PTB$^+$ is a regret minimizer. Indeed, the proof of Proposition 3.1 can be adapted to relate the regret in $\boldsymbol{x}_1, ..., \boldsymbol{x}_T$ in $\mathcal{T}$ to regret in $\boldsymbol{R}_1, ..., \boldsymbol{R}_T$ in $\mathcal{C}_{\geq}$.

**Proposition 4.6.** *Let $\boldsymbol{x}_1, ..., \boldsymbol{x}_T$ be computed by* Smooth PTB$^+$. *Let $\eta = \frac{\sqrt{2\Omega}}{\sqrt{\sum_{t=1}^{T} \|f(\boldsymbol{x}_t, \boldsymbol{\ell}_t) - \boldsymbol{m}_t\|_2^2}}$.*

*Then $\max_{\hat{\boldsymbol{x}} \in \mathcal{T}} \sum_{t=1}^{T} \langle \boldsymbol{x}_t - \hat{\boldsymbol{x}}, \boldsymbol{\ell}_t \rangle \leq \Omega \sqrt{\sum_{t=1}^{T} \|f(\boldsymbol{x}_t, \boldsymbol{\ell}_t) - \boldsymbol{m}_t\|_2^2}$.*

In Smooth PTB⁺ $\boldsymbol{R}_t$ and $\hat{\boldsymbol{R}}_t$ always belong to $\mathcal{C}_{\geq}$, and we are able to recover a RVU bound and show faster convergence. We let $\|\boldsymbol{M}\|$ be the maximum $\ell_2$-norm of any column and any row of $\boldsymbol{M}$.

**Theorem 4.7.** *Let $(\boldsymbol{x}_t)_{t\geq 1}$ and $(\boldsymbol{y}_t)_{t\geq 1}$ be the sequence of strategies computed by both players employing* Smooth PTB⁺ *in the self-play framework, with previous losses as predictions:* $\boldsymbol{m}_t^x = \boldsymbol{f}(\boldsymbol{x}_{t-1}, \boldsymbol{M}\boldsymbol{y}_{t-1}), \boldsymbol{m}_t^y = \boldsymbol{f}(\boldsymbol{y}_{t-1}, -\boldsymbol{M}^\top \boldsymbol{x}_{t-1})$. *Let* $\eta = \frac{R_0}{\sqrt{8d\hat{\Omega}^3 \|\boldsymbol{M}\|}}$ *and* $(\bar{\boldsymbol{x}}_T, \bar{\boldsymbol{y}}_T) = \frac{1}{T}\sum_{t=1}^{T}(\boldsymbol{x}_t, \boldsymbol{y}_t)$. *Then* $\max_{\boldsymbol{y}\in\mathcal{Y}} \langle \bar{\boldsymbol{x}}_T, \boldsymbol{M}\boldsymbol{y}\rangle - \min_{\boldsymbol{x}\in\mathcal{X}} \langle \boldsymbol{x}, \boldsymbol{M}\bar{\boldsymbol{y}}_T\rangle \leq \frac{2\hat{\Omega}^2}{\eta}\frac{1}{T}$.

We present the proof of Theorem 4.7 in Appendix I. To the best of our knowledge, Smooth PTB⁺ is the first algorithm based on Blackwell approachability achieving a $O(1/T)$ convergence rate for solving EFGs as in (1), answering an important open question. However, achieving the faster rate in Smooth PTB⁺ requires introducing a stepsize, a situation similar to all other $O(1/T)$-methods for EFGs, like Mirror Prox and Excessive Gap Technique for EFGs [27] and predictive OMD directly on the treeplex [14]. We can compare the $O(1/T)$ convergence rate of Smooth PTB⁺ with the $O(1/\sqrt{T})$ convergence rate of Predictive CFR⁺ [16], which combines CFR with Predictive RM⁺ (see Appendix B). Despite its predictive nature, Predictive CFR⁺ only achieves a $O(1/\sqrt{T})$ convergence rate because of the CFR decomposition, which enables running regret minimizers *independently and locally* at each infoset, and it is not clear how to combine, at the treeplex level, the regret bounds obtained at each infoset. Since Smooth PTB⁺ operates over the entire treeplex, we can combine the RVU bound for each player to obtain a $O(1/T)$ convergence rate.

**Remark 4.8.** *The Clairvoyant CFR algorithm from [18] is based on Blackwell approachability over* simplexes*, combined with the CFR decomposition and a Mirror Prox-style update [31]. For solving EFGs, Clairvoyant CFR achieves a $O(\log(T)/T))$ convergence rate, slower than the $O(1/T)$ convergence rate of* Smooth PTB⁺*, where the additional $\log(T)$ factor occurs because each outer iteration of Clairvoyant CFR itself solves an approximate fixed-point problem.*

For completeness, we also instantiate Algorithm 1 with regret minimizers that learn heterogeneous stepsizes across information sets in an adaptive fashion. This results in AdaGradTB⁺(Algorithm 6) and AdamTB⁺(Algorithm 7), which adapt the scale of the stepsizes for each dimension to the magnitude of the observed gradients for this dimension based on AdaGrad [12] and Adam [25]. This may be useful if the losses across different dimensions differ in magnitudes, but the stepsizes decrease over time, which could be conservative. These algorithms are presented in Appendix J

**Remark 4.9** (Comparison with Lagrangian Hedging)**.** *Algorithm 1 is related to* Lagrangian Hedging *[21, 11]. Lagrangian Hedging builds upon Blackwell approachability with various potential functions to construct regret minimizers for* general *decision sets. As explained in the introduction, the main focus of our paper is on two-player zero-sum EFGs, i.e., on the case where the decision sets are treeplexes, and where we can obtain several additional interesting properties not studied in [21, 11], such as stepsize invariance, fast convergence rates, and efficient projection, as we detail in the next section. If one were to instantiate Algorithm 1 with the Follow-The-Regularized Leader algorithm, it would yield the regret minimizer for treeplexes studied in Gordon [21], and our Proposition 4.4 in the next section yields an efficient projection oracle for the setup in Gordon [21], which appealed to general convex optimization as an oracle.*

# 5  Numerical Experiments

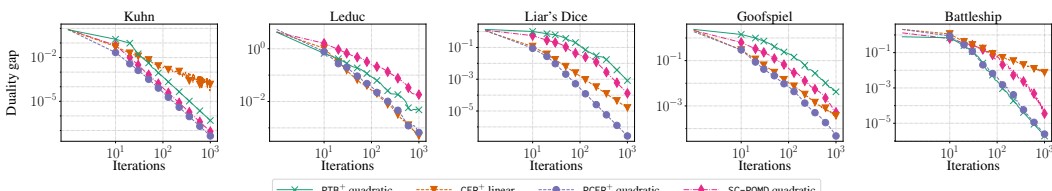

Figure 2: Convergence to a Nash equilibrium for PTB⁺, CFR⁺, PCFR⁺ and SC-POMD. All algorithms use alternation and quadratic averaging except CFR⁺ instantiated with linear averaging.

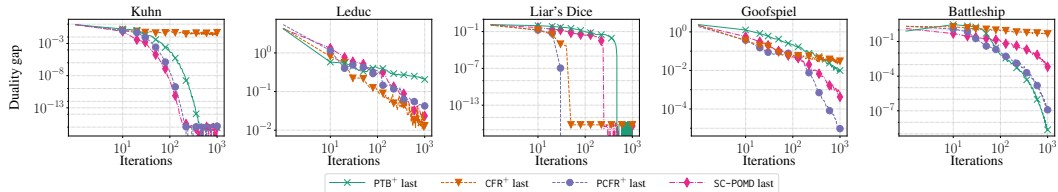

Figure 3: Convergence to a Nash equilibrium for the last iterates of PTB⁺, CFR⁺, PCFR⁺, and SC-POMD. Every algorithm is using alternation.

We conduct two sets of numerical experiments to investigate the performance of our algorithms for solving several two-player zero-sum EFG benchmark games: Kuhn poker, Leduc poker, Liar's Dice, Goofspiel and Battleship. Additional experimental detail is given in Appendix K.

We first determine the best instantiatons our framework. We compare PTB⁺, Smooth PTB⁺ and AdaGradTB⁺ in the self-play framework with alternation (see Appendix A) and uniform, linear or quadratic weights for the iterates. PTB⁺ and Smooth PTB⁺ use the previous losses as predictions. We also study *Treeplex Blackwell*$^+$ (TB⁺), corresponding to PTB⁺ without predictions ($m_t = 0$), and AdamTB⁺. For conciseness, we present our plots in Appendix K.3 (Figure 4) and state our conclusion here. We find that, for every game, PTB⁺ performs the best or is among the best algorithms. This underlines the advantage of *treeplex stepsize invariance* over algorithms that require tuning a stepsize (Smooth PTB⁺) and adaptive algorithms (AdaGradTB⁺), which may perform poorly due to the stepsize decreasing at a rate of $O(1/\sqrt{T})$. AdamTB⁺ does not even converge in some games.

We then compare the best of our algorithms (PTB⁺) with some of the best existing methods for solving EFGs: CFR⁺ [37], predictive CFR⁺ (PCFR⁺, [16], see Appendix B), and a version of optimistic online mirror descent with a single call to the orthogonal projection at every iteration (SC-POMD, [24]) achieving a $O(1/T)$ convergence rate; there are a variety of FOMs with a $O(1/T)$ rate, SC-POMD was observed to perform well in [8]. We determine the best empirical setup for each algorithm in Appendix K.4. In Figure 2, we compare the performance of the (weighted) average iterates. We find that PCFR⁺ outperforms both CFR⁺ and the theoretically-faster SC-POMD, as expected from past work. We had hoped to see at least comparable performance between PTB⁺ and PCFR⁺, since they are both based on Blackwell-approachability regret minimizers derived from applying POMD on the conic hull of their respective decision sets (simplexes at each infoset for PCFR⁺, treeplexes of each player for PTB⁺). However, in some games PCFR⁺ performs much better than PTB⁺. Given the similarity between PTB⁺ and PCFR⁺, our results suggest that the use of the CFR decomposition is part of the key to the performance of PCFR⁺. The CFR decomposition allows PCFR⁺ to have stepsize invariance *at an infoset level*, as opposed to stepsize invariance at the treeplex level in PTB⁺. Because of the structure of treeplexes, the numerical values of variables associated with infosets appearing late in the game, i.e., deeper in the treeplexes, may be much smaller than the numerical values of the variables appearing closer to the root. For this reason, allowing for different stepsizes at each infosets (like CFR⁺ and PCFR⁺ do) appears to be more efficient than using a single stepsize across all the infosets, even when the iterates do not depend on the value of this single stepsize (like in PTB⁺) and when this stepsize is fine-tuned (like in SC-POMD). Of course one could try to run SC-POMD with different stepsizes at each infoset and attempt to tune each of these stepsizes, but this is impossible in practical instances where the number of actions is large, e.g., $4.9 \times 10^4$ actions in *Liar's Dice* and $5.3 \times 10^6$ actions in *Goofspiel*. CFR⁺ and PCFR⁺ bypass this issue with their infoset stepsize invariance, which enables both each infoset to have its own stepsize (via the CFR decomposition) *and* not needing to choose these stepsizes (via using RM⁺ and PRM⁺ as local regret minimizers, which are stepsize invariant).

We also investigate the performance of the *last iterates* in Figure 3. No algorithm appears to be the best across all game instances. CFR⁺ may not converge to a Nash equilibrium (e.g., on Kuhn), as has been observed before [29]. PCFR⁺ exhibits linear convergence in some games (Kuhn, Liar's Dice, Goofspiel) but not others (Leduc). The same is true for PTB⁺. Further investigations about last-iterate convergence are left as an important open question.

# 6 Conclusion

We propose the first Blackwell approachability-based regret minimizer over the treeplex (Algorithm 1) and we give several instantiations of our framework with different properties, including treeplex stepsize invariance (PTB$^+$), adaptive stepsizes (AdaGradTB$^+$) and achieving $O(1/T)$ convergence rates on EFGs with a Blackwell approachability-based algorithm for the first time (Smooth PTB$^+$). Since CFR$^+$ and PCFR$^+$ are stepsize invariant and have strong empirical performance, we were expecting PTB$^+$ to have comparable performance. However, our experiments show that PTB$^+$ often converges slower than CFR$^+$ and PCFR$^+$, so this treeplex stepsize invariance is not the only driver behind the practical performance of CFR$^+$ and PCFR$^+$. We view this negative result as an important contribution of our paper, since it rules out a previously plausible explanation for the practical performance of CFR$^+$. Instead, we propose that one piece of the puzzle behind the CFR$^+$ and PCFR$^+$ performances is their *infoset* stepsize invariance, a consequence of combining the CFR framework with Blackwell approachability-based regret minimizers (RM$^+$ and PRM$^+$, themselves stepsize invariant over simplexes). Future directions include better understanding the last-iterate performance of algorithms based on Blackwell approachability as well as the role of alternation. It would also be interesting to explore EFG applications of new reductions between Blackwell approachability and regret minimization [10] (which differs from the reduction in [2]) and Blackwell approachability generalizations based on various norms and pseudo-norms [28, 9], potentially to obtain better stepsize invariance properties.

## Acknowledgments and Disclosure of Funding

Darshan Chakrabarti was supported by the National Science Foundation Graduate Research Fellowship Program under award number DGE-2036197. Julien Grand-Clément was supported by Hi! Paris and Agence Nationale de la Recherche (Grant 11-LABX-0047). Christian Kroer was supported by the Office of Naval Research awards N00014-22-1-2530 and N00014-23-1-2374, and the National Science Foundation awards IIS-2147361 and IIS-2238960.

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

# A  Self-Play Framework

The (vanilla) self-play framework for two-player zero-sum EFGs is presented in Algorithm 4. The

---
**Algorithm 4** self-play framework
---
1: **Input**: $\mathsf{Regmin}_{\mathcal{X}}$ a regret minimizer over $\mathcal{X}$, $\mathsf{Regmin}_{\mathcal{Y}}$ a regret minimizer over $\mathcal{Y}$
2: **for** $t = 1, \ldots, T$ **do**
3:     $\boldsymbol{x}_t = \mathsf{Regmin}_{\mathcal{X}}(\cdot)$
4:     $\boldsymbol{y}_t = \mathsf{Regmin}_{\mathcal{Y}}(\cdot)$
5:     The first player observes the loss vector $\boldsymbol{A}\boldsymbol{y}_t \in \mathbb{R}^{n_1+1}$
6:     The second player observes the loss vector $-\boldsymbol{A}^\top \boldsymbol{x}_t \in \mathbb{R}^{n_2+1}$
---

self-play framework can be combined with *alternation*, a simple variant that is known to lead to significant empirical speedups, for instance, when CFR⁺ and predictive CFR⁺ are used as regret minimizers [37, 16, 7]. When using alternation, at iteration $t$ the second player is provided with the current strategy of the first player $\boldsymbol{x}_t$ before choosing its own strategy. We describe the self-play framework with alternation in Algorithm 5.

---
**Algorithm 5** self-play framework with alternation
---
1: **Input**: $\mathsf{Regmin}_{\mathcal{X}}$ a regret minimizer over $\mathcal{X}$, $\mathsf{Regmin}_{\mathcal{Y}}$ a regret minimizer over $\mathcal{Y}$
2: **for** $t = 1, \ldots, T$ **do**
3:     $\boldsymbol{x}_t = \mathsf{Regmin}_{\mathcal{X}}(\cdot)$
4:     The second player observes the loss vector $-\boldsymbol{A}^\top \boldsymbol{x}_t \in \mathbb{R}^{n_2+1}$
5:     $\boldsymbol{y}_t = \mathsf{Regmin}_{\mathcal{Y}}(\cdot)$
6:     The first player observes the loss vector $\boldsymbol{A}\boldsymbol{y}_t \in \mathbb{R}^{n_1+1}$
---

# B  Counterfactual Regret Minimization (CFR), CFR⁺ and Predictive CFR⁺

Counterfactual Regret Minimization (CFR, [39]) is a framework for regret minimization over the treeplex. CFR runs a regret minimizer $\mathsf{Regmin}_j$ locally at each infoset $j \in \mathcal{J}$ of the treeplex. Note that here $\mathsf{Regmin}_j$ is a regret minimizer over the *simplex* $\Delta^{n_j}$ with $n_j = |\mathcal{A}_j|$, i.e., over the set of probability distributions over $\mathcal{A}_j$, the set of actions available at infoset $j \in \mathcal{J}$. Let $\boldsymbol{x}_t^j \in \Delta^{n_j}$ be the decision chosen by $\mathsf{Regmin}_j$ at iteration $t$ in CFR and let $\boldsymbol{\ell}_t \in \mathbb{R}^{n+1}$ be the loss across the entire treeplex. The *local loss* $\boldsymbol{\ell}_t^j \in \mathbb{R}^{n_j}$ that CFR passes to $\mathsf{Regmin}_j$ is

$$\ell_{t,a}^j := \ell_{t,(j,a)} + \sum_{j' \in \mathcal{C}_{ja}} V_t^{j'}, \forall\, a \in \mathcal{A}_j, \forall\, j \in \mathcal{J}$$

where $V_t^j$ is the *value function* for infoset $j$ at iteration $t$, defined inductively:

$$V_t^j := \sum_{a \in \mathcal{A}_j} x_{t,a}^j \ell_{t,(j,a)} + \sum_{j' \in \mathcal{C}_{ja}} V_t^{j'}.$$

The regret over the entire treeplex $\mathcal{T}$ can be related to the regrets accumulated at each infoset via the following *laminar regret decomposition* [13]:

$$\mathsf{Reg}^T := \max_{\hat{\boldsymbol{x}} \in \mathcal{T}} \sum_{t=1}^T \langle \boldsymbol{x}_t - \hat{\boldsymbol{x}}, \boldsymbol{\ell}_t \rangle \le \max_{\hat{\boldsymbol{x}} \in \mathcal{T}} \sum_{j \in \mathcal{J}} \hat{x}_{p_j} \mathsf{Reg}_j^T\left(\hat{\boldsymbol{x}}^j\right)$$

with $\mathsf{Reg}_j^T\left(\hat{\boldsymbol{x}}^j\right) := \sum_{t=1}^T \langle \boldsymbol{x}_t^j - \hat{\boldsymbol{x}}^j, \boldsymbol{\ell}_t^j \rangle$ the regret incured by $\mathsf{Regmin}^j$ for the sequence of losses $\boldsymbol{\ell}_1^j, \ldots, \boldsymbol{\ell}_T^j$ against the comparator $\hat{\boldsymbol{x}}^j \in \Delta^{n_j}$. Combining CFR with regret minimizers at each information set ensures $\mathsf{Reg}^T = O\left(\sqrt{T}\right)$.

CFR$^+$ [37] corresponds to instantiating the self-play framework with alternation (Algorithm 5) and Regret Matching$^+$ (RM$^+$ as presented in (4)) as a regret minimizer at each information set. Additionally, CFR$^+$ uses *linear* averaging, i.e., it returns $\bar{\boldsymbol{x}}_T$ such that $\bar{\boldsymbol{x}}_T = \frac{1}{\sum_{t=1}^T \omega_t} \sum_{t=1}^T \omega_t \boldsymbol{x}_t$ with $\omega_t = t$. We also consider uniform weights ($\omega_t = 1$) and quadratic weights ($\omega = t^2$) in our simulations (Figure 10). CFR$^+$ guarantees a $O(1/\sqrt{T})$ convergence rate to a Nash equilibrium.

Predictive CFR$^+$(PCFR$^+$, [16]) corresponds to instantiating the self-play framework with alternation (Algorithm 5) and Predictive Regret Matching$^+$ (PRM$^+$) as a regret minimizer at each information set. Given a simplex $\Delta^d$, PRM$^+$ is a regret minimizer that returns a sequence of decisions $\boldsymbol{z}_1, ..., \boldsymbol{z}_T \in \Delta^d$ as follows:

$$\hat{\boldsymbol{R}}_t = \Pi_{\mathbb{R}_+^d} \left( \boldsymbol{R}_t - \eta \boldsymbol{g}(\boldsymbol{z}_{t-1}, \boldsymbol{\ell}_{t-1}) \right)$$

$$\boldsymbol{z}_t = \hat{\boldsymbol{R}}_t / \|\hat{\boldsymbol{R}}_t\|_1, \qquad \text{(PRM}^+\text{)}$$

$$\boldsymbol{R}_{t+1} = \Pi_{\mathbb{R}_+^d} \left( \boldsymbol{R}_t - \eta \boldsymbol{g}(\boldsymbol{z}_t, \boldsymbol{\ell}_t) \right)$$

where the function $g$ is defined in (5). Similar to CFR$^+$, for PCFR$^+$ we investigate different weighting schemes in our numerical experiments (Figure 11). It is not known if the self-play framework with alternation, combined with PCFR$^+$, has convergence guarantees, but PCFR$^+$ has been observed to achieve state-of-the-art practical performance in many EFG instances [16].

# C  Comparison with [1]

In this appendix, we describe the results in [1] connecting Blackwell approachability and regret minimization, and we highlight the main differences with our framework described in Algorithm 1.

In particular, Abernethy et al. [1] describe a meta-algorithm connecting Blackwell approachability and regret minimization. Given a decision set $\mathcal{X} \subset \mathbb{R}^n$ assumed to be convex and compact, Abernethy et al. [1] consider a *lifted set* $\tilde{\mathcal{X}} = \{\kappa\} \times \mathcal{X} \subset \mathbb{R}^{n+1}$ with

$$\kappa := \max_{\boldsymbol{x} \in \mathcal{X}} \|\boldsymbol{x}\|_2.$$

Abernethy et al. [1] then constructs a regret minimizer over $\mathcal{X}$ by considering a Blackwell approachability instance, where the target set is defined as $\text{cone}(\tilde{\mathcal{X}}) \subset \mathbb{R}^{n+1}$, the decision set is $\mathcal{X} \subset \mathbb{R}^n$, and the instantaneous loss at time $t$ is $\left( \frac{\langle \boldsymbol{x}_t, \boldsymbol{\ell}_t \rangle}{\kappa}, -\boldsymbol{\ell}_t \right) \in \mathbb{R}^{n+1}$ with $\boldsymbol{\ell}_t \in \mathbb{R}^n$ the loss vector when the decision maker chooses $\boldsymbol{x}_t \in \mathcal{X}$. The *average aggregated loss vector* $\boldsymbol{u}_t \in \mathbb{R}^{n+1}$ is updated using a regret minimizer over $\text{cone}(\tilde{\mathcal{X}})$ and the decision maker in the Blackwell approachability instance chooses the sequence of decisions $\boldsymbol{x}_1, ..., \boldsymbol{x}_T$ to ensure that $\left( \frac{1}{T} \boldsymbol{u}_T \right)_{T \geq 1}$ approaches the *target set*, defined as the polar cone of $\text{cone}(\tilde{\mathcal{X}})$. We refer to Section 4 in [1] for more detail on this construction.

As evident from the description in the previous paragraph, our framework described in Algorithm 1 differs from the meta-algorithm from [1] in various ways. In particular, if one were to directly use the reduction from [1] for deriving regret minimizers over treeplexes, one would need to consider $\tilde{\mathcal{T}} = \{\kappa\} \times \mathcal{T}$, with $\mathcal{T}$ a treeplex and $\kappa = \max_{\boldsymbol{x} \in \mathcal{T}} \|\boldsymbol{x}\|_2$, and one would need to consider a bound for the value of $\kappa$, which could be conservative for large-scale EFG instances. However, since we have designed Algorithm 1 specifically for treeplexes as decision sets, we do *not* need to lift the set $\mathcal{T}$ by adding an additional dimension depending on the maximum $\ell_2$-norm over $\mathcal{T}$. We can circumvent relying on $\kappa$ and therefore obtain a simpler, more practical framework as in Algorithm 1 by exploiting the structure of treeplexes. This is because the variable $x_\varnothing$ associated with the empty sequence always has a value of 1: $x_\varnothing = 1$, and we can then exploit the fact that $\mathcal{T} \subset \{\boldsymbol{x} \in \mathbb{R}^{n+1} \mid \langle \boldsymbol{x}, \boldsymbol{a} \rangle = 1\}$, as in the proof of Proposition 3.1.

Another fundamental difference with [1] is our positioning and our objectives. Abernethy et al. [1] analyze Blackwell approachability with *adversarial losses* and in a more theoretical way (e.g., no implementations or simulations), whereas we focus on practically solving EFGs with Blackwell approachability, i.e., we focus on the game setting and on explaining the empirical performance of CFR$^+$. A direct application of the results in [1] would only result in algorithms achieving $O(1/\sqrt{T})$ convergence rates, and for which no concrete implementations are known. In contrast, we provide

details on the practical implementations of our algorithms (Proposition 4.4 and Appendix H), we are the first to highlight the role of stepsize invariance, which only makes sense for EFGs (and not for the adversarial loss setup as in [1]), and in our EFG applications we can obtain faster $O(1/T)$ convergence rate (e.g. for Smooth PTB$^+$, see our new results as in Proposition 4.5 and Theorem 4.7), which is impossible against adversarial losses.

# D   Proof of Proposition 4.1

*Proof.* The proof of Proposition 4.1 is based on the following lemma.

**Lemma D.1.** *Let $\mathcal{C} \subset \mathbb{R}^n$ be a convex cone and let $\boldsymbol{u} \in \mathbb{R}^n, \eta > 0$. Then*
$$\Pi_{\mathcal{C}}(\eta\boldsymbol{u}) = \eta\Pi_{\mathcal{C}}(\boldsymbol{u}).$$

*Proof of Lemma D.1.* We have, by definition,
$$\Pi_{\mathcal{C}}(\eta\boldsymbol{u}) = \arg\min_{\boldsymbol{R}\in\mathcal{C}} \|\boldsymbol{R} - \eta\boldsymbol{u}\|_2.$$

Now we also have
$$\min_{\boldsymbol{R}\in\mathcal{C}} \|\boldsymbol{R} - \eta\boldsymbol{u}\|_2 = \eta \cdot \min_{\boldsymbol{R}\in\mathcal{C}} \|\frac{1}{\eta}\boldsymbol{R} - \boldsymbol{u}\|_2 = \eta \cdot \min_{\boldsymbol{R}\in\mathcal{C}} \|\boldsymbol{R} - \boldsymbol{u}\|_2$$
where the last equality follows from $\mathcal{C}$ being a cone. This shows that $\arg\min_{\boldsymbol{R}\in\mathcal{C}} \|\boldsymbol{R} - \eta\boldsymbol{u}\|_2$ is attained at $\eta\Pi_{\mathcal{C}}(\boldsymbol{u})$, i.e., that $\Pi_{\mathcal{C}}(\eta\boldsymbol{u}) = \eta\Pi_{\mathcal{C}}(\boldsymbol{u})$. $\qquad\square$

We are now ready to prove Proposition 4.1. For the sake of conciseness we prove this with $\boldsymbol{m}_1 = ... = \boldsymbol{m}_T = \boldsymbol{0}$; the proof for PTB$^+$ with predictions is identical. In this case, $\boldsymbol{R}_t = \hat{\boldsymbol{R}}_t, \forall\ t \geq 1$. Consider the sequence of strategies $\tilde{\boldsymbol{x}}_1, ..., \tilde{\boldsymbol{x}}_T$ and $\tilde{\boldsymbol{R}}_1, ..., \tilde{\boldsymbol{R}}_T$ generated by PTB$^+$ with a step size of 1. We also consider the sequence of strategies $\boldsymbol{x}_1, ..., \boldsymbol{x}_T$ and $\boldsymbol{R}_1, ..., \boldsymbol{R}_T$ generated with a step size $\eta > 0$. We claim that
$$\tilde{\boldsymbol{x}}_t = \boldsymbol{x}_t, \boldsymbol{R}_t = \eta\tilde{\boldsymbol{R}}_t, \ \forall\ t \in \{1, ..., T\}.$$
We prove this by induction. Both sequences of iterates are initialized with $\boldsymbol{R}_1 = \tilde{\boldsymbol{R}}_1 = \boldsymbol{0}$ so that $\tilde{\boldsymbol{x}}_1 = \boldsymbol{x}_1$. Therefore, both sequences face the same loss $\boldsymbol{\ell}_1$ at $t = 1$, and we have
$$\boldsymbol{R}_2 = \Pi_{\mathcal{C}}(-\eta\boldsymbol{f}(\boldsymbol{x}_1, \boldsymbol{\ell}_1)) = \eta\pi_{\mathcal{C}}(-\boldsymbol{f}(\boldsymbol{x}_1, \boldsymbol{\ell}_1))) = \eta\tilde{\boldsymbol{R}}_2.$$
Let us now consider an iteration $t \geq 1$ and suppose that $\tilde{\boldsymbol{x}}_t = \boldsymbol{x}_t, \boldsymbol{R}_t = \eta\tilde{\boldsymbol{R}}_t$. Since $\tilde{\boldsymbol{x}}_t = \boldsymbol{x}_t$ then both algorithms will face the next loss vector $\boldsymbol{\ell}_t$. Then
$$\begin{aligned}
\boldsymbol{R}_{t+1} &= \pi_{\mathcal{C}}(\boldsymbol{R}_t - \eta\boldsymbol{f}(\boldsymbol{x}_t, \boldsymbol{\ell}_t)) \\
&= \pi_{\mathcal{C}}(\eta\tilde{\boldsymbol{R}}_t - \eta\boldsymbol{f}(\boldsymbol{x}_t, \boldsymbol{\ell}_t)) \\
&= \eta\pi_{\mathcal{C}}(\tilde{\boldsymbol{R}}_t - \boldsymbol{f}(\boldsymbol{x}_t, \boldsymbol{\ell}_t)) \\
&= \eta\tilde{\boldsymbol{R}}_{t+1}
\end{aligned}$$
which in turns implies that $\boldsymbol{x}_{t+1} = \tilde{\boldsymbol{x}}_{t+1}$. We conclude that $\boldsymbol{x}_t = \tilde{\boldsymbol{x}}_t, \forall\ t = 1, ..., T$. $\qquad\square$

# E   Comparison Between RM$^+$ and PTB$^+$

For the sake of discussion, we assume that the original decision set of each player is a simplex $\Delta^d$ and that there are no predictions: $\boldsymbol{m}_t = \boldsymbol{0}, \forall\ t \geq 1$.

PTB$^+$ **over the simplex.**   For PTB$^+$, the empty sequence variable $x_\varnothing$ is introduced and appended to the decision $\Delta^d$. The resulting treeplex can be written $\mathcal{T} = \{1\} \times \Delta^d$, the set $\mathcal{C}$ becomes $\mathcal{C} := \text{cone}(\mathcal{T}) = \text{cone}(\{1\} \times \Delta^d)$ and $\boldsymbol{a} = (1, \boldsymbol{0}) \in \mathbb{R}^{d+1}_+$ with 1 on the first component related to $x_\varnothing$ and 0 everywhere else. In this case, PTB$^+$ without prediction is exactly the *Conic Blackwell Algorithm$^+$* (CBA$^+$, [22]). Crucially, to run PTB$^+$ we need to compute the orthogonal projection onto $\text{cone}(\mathcal{T}) = \text{cone}(\{1\} \times \Delta)$ at every iteration, which can not be computed in closed-form, but it can be computed in $O(n \log(n))$ arithmetic operations (see Appendix G.1 in [22]).

**Regret Matching$^+$.** RM$^+$ operates directly over the simplex $\Delta^d$ without the introduction of the empty sequence $x_\varnothing$, in contrast to PTB$^+$ which operates over $\{1\} \times \Delta^d$. Importantly, in RM$^+$, at every iteration the orthogonal projection onto $\mathbb{R}^d_+$ can be computed in closed form by simply thresholding to zero the negative components (and leaving unchanged the positive components): $\Pi_{\mathbb{R}^d_+}(\boldsymbol{z}) = (\max\{z_i, 0\})_{i \in [d]}$ for any $\boldsymbol{z} \in \mathbb{R}^d$.

**Empirical comparisons over simplexes.** The numerical experiments in [22] show that CBA$^+$ may be slightly faster than RM$^+$ for some matrix games in terms of speed of convergence as a function of the number of iterations, but it can be slower in running times because of the orthogonal projections onto cone($\{1\} \times \Delta$) at each iteration (Figures 2,3,4 in [22]). When $\mathcal{T}$ is a treeplex that is not the simplex, introducing $x_\varnothing$ also changes the resulting algorithm but not the complexity of the orthogonal projection onto cone($\mathcal{T}$), since there is no closed-form anymore, even without $x_\varnothing$. As a convention, in this paper, we will always use $x_\varnothing$ in our description of treeplexes and of our algorithms since it is convenient from a writing and implementation standpoint.

Overall, we notice that in the case of the simplex introducing the empty sequence variable $x_\varnothing$ radically alters the complexity per iterations and the resulting algorithm, a fact that has not been noticed in previous work.

**Empirical comparisons for EFGs.** For solving EFGs, [22] combine the CFR decomposition with CBA$^+$ and compare the resulting algorithm with CFR$^+$ (i.e., combining the CFR decomposition with RM$^+$). The authors in [22] observe similar numerical results as for the cases of simplexes: the resulting algorithm may slightly outperform CFR$^+$ in terms of duality gap achieved after a certain number of iterations, but it is outperformed by CFR$^+$ in terms of duality gap achieved after a certain computation time, because of the orthogonal projection required at every iteration at every simplex present in the treeplexes of each player.

# F   Proof of Proposition 4.5

*Proof of Proposition 4.5.*   1. Let $\hat{\boldsymbol{R}}_2 = \boldsymbol{R}_2/\|\boldsymbol{R}_2\|_2$ be the unit vector pointing in the same direction as $\boldsymbol{R}_2$ and let $\boldsymbol{h} := \left(\langle \boldsymbol{R}_1, \hat{\boldsymbol{R}}_2 \rangle\right) \hat{\boldsymbol{R}}_2$ the orthogonal projection of $\boldsymbol{R}_1$ onto $\{\alpha \hat{\boldsymbol{R}}_2 \mid \alpha \in \mathbb{R}\}$. We thus have $\|\boldsymbol{R}_1 - \boldsymbol{R}_2\|_2 \geq \|\boldsymbol{R}_1 - \boldsymbol{h}\|_2$.

2. Let $\boldsymbol{p} = \frac{\langle \boldsymbol{R}_1, \boldsymbol{a} \rangle}{\langle \boldsymbol{R}_2, \boldsymbol{a} \rangle} \hat{\boldsymbol{R}}_2$. Since $\boldsymbol{p}$ and $\boldsymbol{R}_2$ are colinear, we have

$$\left\| \frac{\boldsymbol{R}_1}{\langle \boldsymbol{R}_1, \boldsymbol{a} \rangle} - \frac{\boldsymbol{R}_2}{\langle \boldsymbol{R}_2, \boldsymbol{a} \rangle} \right\|_2 = \left\| \frac{\boldsymbol{R}_1}{\langle \boldsymbol{R}_1, \boldsymbol{a} \rangle} - \frac{\boldsymbol{p}}{\langle \boldsymbol{p}, \boldsymbol{a} \rangle} \right\|_2.$$

Additionally, by construction, $\langle \boldsymbol{p}, \boldsymbol{a} \rangle = \langle \boldsymbol{R}_1, \boldsymbol{a} \rangle$, so that we obtain

$$\left\| \frac{\boldsymbol{R}_1}{\langle \boldsymbol{R}_1, \boldsymbol{a} \rangle} - \frac{\boldsymbol{R}_2}{\langle \boldsymbol{R}_2, \boldsymbol{a} \rangle} \right\|_2 = \left\| \frac{\boldsymbol{R}_1}{\langle \boldsymbol{R}_1, \boldsymbol{a} \rangle} - \frac{\boldsymbol{p}}{\langle \boldsymbol{R}_1, \boldsymbol{a} \rangle} \right\|_2 = \frac{1}{\langle \boldsymbol{R}_1, \boldsymbol{a} \rangle} \|\boldsymbol{R}_1 - \boldsymbol{p}\|_2.$$

Note that $\langle \boldsymbol{R}_1, \boldsymbol{a} \rangle \geq 0$ since $\boldsymbol{R}_1 \in \mathrm{cone}(\mathcal{T})$ and $\mathcal{T} \subset \{\boldsymbol{x} \in \mathbb{R}^{n+1} \mid \langle \boldsymbol{x}, \boldsymbol{a} \rangle = 1\}$. Assume that we can compute $D > 0$ such that $\frac{\|\boldsymbol{R}_1 - \boldsymbol{p}\|_2}{\|\boldsymbol{R}_1 - \boldsymbol{h}\|_2} \leq D$. Then we have

$$\left\| \frac{\boldsymbol{R}_1}{\langle \boldsymbol{R}_1, \boldsymbol{a} \rangle} - \frac{\boldsymbol{R}_2}{\langle \boldsymbol{R}_2, \boldsymbol{a} \rangle} \right\|_2 \leq \frac{D}{\langle \boldsymbol{R}_1, \boldsymbol{a} \rangle} \|\boldsymbol{R}_1 - \boldsymbol{h}\|_2 \leq \frac{D}{\langle \boldsymbol{R}_1, \boldsymbol{a} \rangle} \|\boldsymbol{R}_1 - \boldsymbol{R}_2\|_2.$$

3. The rest of this proof focuses on showing that $\frac{\|\boldsymbol{R}_1 - \boldsymbol{p}\|_2}{\|\boldsymbol{R}_1 - \boldsymbol{h}\|_2} \leq \Omega$ with $\Omega = \max\{\|\boldsymbol{x}\|_2 \mid \boldsymbol{x} \in \mathcal{T}\}$. Note that $\langle \boldsymbol{R}_1 - \boldsymbol{p}, \boldsymbol{a} \rangle = 0$. Therefore, $\frac{1}{\|\boldsymbol{R}_1 - \boldsymbol{p}\|_2}(\boldsymbol{R}_1 - \boldsymbol{p})$ and $\frac{1}{\|\boldsymbol{a}\|_2} \boldsymbol{a}$ can be completed to form an orthonormal basis of $\mathbb{R}^n$. In this basis, we have

$$\|\hat{\boldsymbol{R}}_2\|_2^2 \geq \frac{\left(\langle \boldsymbol{R}_1 - \boldsymbol{p}, \hat{\boldsymbol{R}}_2 \rangle\right)^2}{\|\boldsymbol{R}_1 - \boldsymbol{p}\|_2^2} + \frac{\left(\langle \boldsymbol{a}, \hat{\boldsymbol{R}}_2 \rangle\right)^2}{\|\boldsymbol{a}\|_2^2}.$$

Note that by construction we have $\|\hat{R}_2\|_2^2 = 1$. Additionally, $R_2 \in \text{cone}(\mathcal{T})$ so that there exists $\alpha > 0$ and $y \in \mathcal{T}$ such that $R_2 = \alpha x$. By construction of $\hat{R}_2$, we have $\hat{R}_2 = \frac{\alpha x}{\|\alpha x\|_2} = \frac{x}{\|x\|_2}$ and $\langle x, a \rangle = 1$. This shows that

$$\frac{\left(\langle a, \hat{R}_2 \rangle\right)^2}{\|a\|_2^2} = \frac{(\langle a, x \rangle)^2}{\|a\|_2^2 \|x\|_2^2} = \frac{1}{\|a\|_2^2 \|x\|_2^2} \geq \frac{1}{\Omega \|a\|_2^2}$$

with $\Omega = \max\{\|x\|_2| \ x \in \mathcal{T}\}$. Recall that we have chosen $a = (1, \mathbf{0})$ so that $\|a\|_2 = 1$. Overall, we have obtained

$$1 - \frac{1}{\Omega^2} \geq \frac{\left(\langle R_1 - p, \hat{R}_2 \rangle\right)^2}{\|R_1 - p\|_2^2}.$$

From the definition of the vectors $p, h$ and $\hat{R}_2$, we have

$$\frac{\left(\langle R_1 - p, \hat{R}_2 \rangle\right)^2}{\|R_1 - p\|_2^2} = \frac{\|p - h\|_2^2}{\|R_1 - p\|_2^2}.$$

Hence, we have

$$\|p - h\|_2^2 \leq \left(1 - \frac{1}{\Omega^2}\right) \|R_1 - p\|_2^2.$$

This shows that $\|R_1 - h\|_2^2 \geq \frac{1}{\Omega^2} \|x - p\|_2^2$.

4. We conclude that

$$\left\| \frac{R_1}{\langle R_1, a \rangle} - \frac{R_2}{\langle R_2, a \rangle} \right\|_2 \leq \frac{\Omega}{\max\{\langle R_1, a \rangle, \langle R_2, a \rangle\}} \|R_1 - R_2\|_2.$$

$\square$

# G   Proof of Proposition 4.4

In this section we show how to efficiently compute the orthogonal projection onto the cone $\mathcal{C} := \text{cone}(\mathcal{T})$. We start by reviewing the existing methods for computing the orthogonal projection onto the treeplex $\mathcal{T}$. This is an important cornerstone of our analysis, since the treeplex $\mathcal{T}$ and the cone $\mathcal{C}$ share an analogous structure:

$$\mathcal{T} = \{x \in \mathbb{R}_+^{n+1} \mid x_\varnothing = 1, \sum_{a \in \mathcal{A}_j} x_{ja} = x_{p_j}, \forall j \in \mathcal{J}\}$$

$$\mathcal{C} = \{x \in \mathbb{R}_+^{n+1} \mid \sum_{a \in \mathcal{A}_j} x_{ja} = x_{p_j}, \forall j \in \mathcal{J}\}.$$

[20] were the first to show an algorithm for computing Euclidean projection onto the treeplex. They do this by defining a value function for the projection of a given point $y$ onto the closed and convex *scaled* set $t\mathcal{Z}$, letting it be half the squared distance between $y$ and $t\mathcal{Z}$, for $t \in \mathbb{R}_{>0}$:

$$v_{\mathcal{Z}}(t, y) := \frac{1}{2} \min_{z \in t\mathcal{Z}} \|z - y\|_2^2.$$

[20] show how to recursively compute $\lambda_{\mathcal{Z}}(t, y)$, the derivative of this function with respect to $t$, for a given treeplex, since treeplexes can be constructed recursively using two operations: branching and Cartesian product. In the first case, given $k$ treeplexes $\mathcal{Z}_1, \ldots, \mathcal{Z}_k$, then $\mathcal{Z} = \{x, x[1]z_1, \ldots, x[k]z_k : x \in \Delta_k, z_i \in \mathcal{Z}_i \forall i \in [k]\}$ is also a treeplex. In the second case, given $k$ treeplexes $\mathcal{Z}_1, \ldots, \mathcal{Z}_k$, then $\mathcal{Z} = \mathcal{Z}_1 \times \cdots \times \mathcal{Z}_k$ is also a treeplex. In fact, letting the empty set be a treeplex as a base case, all treeplexes can be constructed in this way.

However, [20] did not state the total complexity of computing the projection, instead only stating the complexity of computing $\lambda_{\mathcal{Z}}(t, y)$ given the corresponding $\lambda_{\mathcal{Z}_i}(t, y_i)$ functions for the treeplexes $\mathcal{Z}_i$

that are used to construct $\mathcal{Z}$ using $i \in [k]$. They state that this complexity is $O(n \log n)$, where $n$ is the number of sequences in $\mathcal{Z}$. Their analysis involves showing that the function $t \mapsto \lambda_{\mathcal{Z}}(t, \boldsymbol{y})$ is piecewise linear.

[17] also consider this problem, generalizing the problem to weighted projection on the scaled treeplex, by adding an additional positive parameter $\boldsymbol{w} \in \mathbb{R}^n_{>0}$:

$$v_{\mathcal{Z}}(t, \boldsymbol{y}, \boldsymbol{w}) := \frac{1}{2} \min_{\boldsymbol{z} \in t\mathcal{Z}} \sum_{i=1}^n \left( \boldsymbol{z}[i] - \frac{\boldsymbol{y}[i]}{\boldsymbol{w}[i]} \right)^2 .$$

They do a similar analysis to [20], by showing how to compute the derivative $\lambda_{\mathcal{Z}}(t, \boldsymbol{y}, \boldsymbol{w})$ of $v_{\mathcal{Z}}(t, \boldsymbol{y}, \boldsymbol{w})$ with respect to $t$ recursively. They show that $t \mapsto \lambda_{\mathcal{Z}}(t, \boldsymbol{y}, \boldsymbol{w})$ are strictly-monotonically-increasing piecewise-linear (SMPL) functions. We will follow the analysis in [17], letting $\boldsymbol{w} = \mathbf{1}$.

We first define a standard representation of a SMPL function.

**Definition G.1** ([17])**.** *Given a SMPL function $f$, a standard representation is an expression of the form*

$$f(x) = \zeta + \alpha_0 x + \sum_{s=1}^S \alpha_s \max\{0, x - \beta_s\}$$

*valid for all $x \in \mathrm{dom}(f)$, $S \in \mathbb{N} \cup \{0\}$, and $\beta_1 < \cdots < \beta_S$. The size of the standard representation is defined to be $S$.*

Next, we prove the following lemma, showing the computational complexity of computing the derivative of the value function for a given treeplex.

**Lemma G.2.** *For a given treeplex $\mathcal{Z}$ with depth $d$, $n$ sequences, $l$ leaf sequences, and $m$ infosets, and $\boldsymbol{y} \in \mathbb{R}^n, \boldsymbol{w} \in \mathbb{R}^n_{>0}$, a standard representation of $\lambda_{\mathcal{Z}}(t, \boldsymbol{y}, \boldsymbol{w})$ can be computed in $O\big(dn \log(l + m)\big)$ time.*

*Proof.* We will proceed by structural induction over treeplexes, following the analysis done by [17]. The base case is trivially true, because the empty set has no sequences or depth.

For the inductive case, we will assume that it requires $O\big((d - 1)n \log(l + m)\big)$ time to compute the respective Euclidean projections onto the subtreeplexes that we use to inductively construct our current treeplex, where $d - 1$ is the depth of a given subtreeplex, $n$ is the number of sequences in the subtreeplex, and $m$ is the total number of sequences among both players and chance corresponding to the game from which the treeplex originates.

We will use two results shown in Lemma 14 of [17]:

**Lemma G.3** (Recursive complexity of Euclidean projection for branching operation [17])**.** *Consider a treeplex $\mathcal{Z}$ that can be written as the result of a branching operation on $k$ treeplexes $\mathcal{Z}_1, \ldots, \mathcal{Z}_k$:*

$$\mathcal{Z} = \{\boldsymbol{x}, \boldsymbol{x}[1]\boldsymbol{z}_1, \ldots, \boldsymbol{x}[k]\boldsymbol{z}_k : x \in \Delta_k, \boldsymbol{z}_i \in \mathcal{Z}_i \forall i \in [k]\}.$$

*Let $\mathcal{Z}$ have $n$ sequences and let $\boldsymbol{y}, \boldsymbol{w} \in \mathbb{R}^n$, and let $\boldsymbol{y}[i]$ and $\boldsymbol{w}[i]$ denote the corresponding respective components of $\boldsymbol{y}$ and $\boldsymbol{w}$ for the treeplex $\mathcal{Z}_i$.*

*Then, given standard representations of $\lambda_{\mathcal{Z}_i}(t, \boldsymbol{y}_i, \boldsymbol{w}_i)$ of size $n_i$ for all $i \in [k]$, where $n_i$ is the number of sequences that $\mathcal{Z}_i$ has, a standard representation of $\lambda_{\mathcal{Z}}(t, \boldsymbol{y}, \boldsymbol{w})$ of size $n$ can be computed in $O(n \log k)$ time.*

*Furthermore, given a value of $t$, the argument $\boldsymbol{x}$ which leads to the realization of the optimal value of the value function, can be computed in time $O(n)$.*

**Lemma G.4** (Recursive complexity of Euclidean projection for Cartesian product [17])**.** *Consider a treeplex $\mathcal{Z}$ that can be written as a Cartesian product of $k$ treeplexes $\mathcal{Z}_1, \ldots, \mathcal{Z}_k$:*

$$\mathcal{Z} = \mathcal{Z}_1 \times \cdots \times \mathcal{Z}_k.$$

*Let $\mathcal{Z}$ have $n$ sequences and let $\boldsymbol{y}, \boldsymbol{w} \in \mathbb{R}^n$, and let $\boldsymbol{y}[i]$ and $\boldsymbol{w}[i]$ denote the corresponding respective components of $\boldsymbol{y}$ and $\boldsymbol{w}$ for the treeplex $\mathcal{Z}_i$.*

*Then, given standard representations of $\lambda_{\mathcal{Z}_i}(t, \boldsymbol{y}_i, \boldsymbol{w}_i)$ of size $n_i$ for all $i \in [k]$, where $n_i$ is the number of sequences that $\mathcal{Z}_i$ has, a standard representation of $\lambda_{\mathcal{Z}}(t, \boldsymbol{y}, \boldsymbol{w})$ of size $n$ can be computed in $O(n \log k)$ time.*

First, we consider the case that the last operation used to construct our treeplex was the branching operation. Let the root of of the treeplex be called $j$. Define $\mathcal{Z}_i$ as the treeplex that is underneath action $a_i \in \mathcal{A}_j$. Let $n_i$ denote the number of sequences in $\mathcal{Z}_i$, $m_i$ denote the number of infosets in $\mathcal{Z}_i$, $l_i$ denote the number of leaf sequences in $\mathcal{Z}_i$, and $d-1$ be the maximum depth of any of these subtreeplexes.

Given a standard representation of $\lambda_{\mathcal{Z}_i}(t, \boldsymbol{y}_i, \boldsymbol{w}_i)$ of size $n_i$ for all $i \in [|\mathcal{A}_j|]$, by Lemma G.3, it takes $O(n \log |\mathcal{A}_j|)$ time to compute a standard representation of $\lambda_{\mathcal{Z}}(t, \boldsymbol{y}, \boldsymbol{w})$ of size $n$. By induction, it takes $O\big((d-1)n_i \log m_i\big)$ to compute $\lambda_{\mathcal{Z}_i}(t, \boldsymbol{y}_i, \boldsymbol{w}_i)$ for treeplex $\mathcal{Z}_i$. Thus the total computation required to compute $\lambda_{\mathcal{Z}}(t, \boldsymbol{y}, \boldsymbol{w})$ is

$$
\begin{aligned}
O(n \log |\mathcal{A}_j|) + \sum_{i \in [|\mathcal{A}_j|]} O\big((d-1)n_i \log(l_i + m_i)\big) &= O(n \log |\mathcal{A}_j|) + \sum_{i \in [|\mathcal{A}_j|]} O\big((d-1)n_i \log(l+m)\big) \\
&= O(n \log |\mathcal{A}_j|) + O\big((d-1) \sum_{i \in [|\mathcal{A}_j|]} n_i \log(l+m)\big) \\
&= O(n \log |\mathcal{A}_j|) + O\big((d-1)n \log(l+m)\big) \\
&= O\big(n \log(l+m)\big) + O\big((d-1)n \log(l+m)\big) \\
&= O\big(dn \log(l+m)\big)
\end{aligned}
$$

since we have necessarily that $l_i \le l$ and $m_i \le m$ for all $i \in [|\mathcal{A}_j|]$, $\sum_{i \in [|\mathcal{A}_j|]} n_i \le n$, and $|\mathcal{A}_j| \le l + m$.

Second, we consider the case the last operation to construct our treeplex was a Cartesian product. Let $\mathcal{Z} = \mathcal{Z}_1 \times \cdots \times \mathcal{Z}_k$, and again define $n_i$ as the number of sequences in $\mathcal{Z}_i$, $m_i$ as the number of infosets in $\mathcal{Z}_i$, $l_i$ as the number of leaf sequences in $\mathcal{Z}_i$, and $d-1$ as the maximum depth of any of these subtreeplexes.

Given a standard representation of $\lambda_{\mathcal{Z}_i}(t, \boldsymbol{y}_i, \boldsymbol{w}_i)$ of size $n_i$ for all $i \in [k]$, by Lemma G.4 it takes $O(n \log k)$ to compute a standard representation of $\lambda_{\mathcal{Z}}(t, \boldsymbol{y}, \boldsymbol{w})$ of size $n$. By induction, it takes $O\big((d-1)n_i \log(l_i + m_i)\big)$ to compute $\lambda_{\mathcal{Z}_i}(t, \boldsymbol{y}_i, \boldsymbol{w}_i)$ for treeplex $\mathcal{Z}_i$. Thus the total computation required to compute $\lambda_{\mathcal{Z}}(t, \boldsymbol{y}, \boldsymbol{w})$ is

$$
\begin{aligned}
O(n \log k) + \sum_{i \in [k]} O\big((d-1)n_i \log(l_i + m_i)\big) &= O(n \log k) + \sum_{i \in [k]} O\big((d-1)n_i \log(l+m)\big) \\
&= O(n \log k) + O\big((d-1) \sum_{i \in [k]} n_i \log(l+m)\big) \\
&= O(n \log k) + O\big((d-1)n \log(l+m)\big) \\
&= O(n \log m) + O\big((d-1)n \log(l+m)\big) \\
&= O(dn \log(l+m))
\end{aligned}
$$

since we have necessarily that $l_i \le l$ and $m_i \le m$ for all $i \in [k]$, and $k \le m$. $\qquad\square$

Finally, we are ready to prove the main statement.

*Proof of Proposition 4.4.* By Lemma G.2, we know that we can recursively compute a standard representation of $\lambda_{\mathcal{Z}}(t, \boldsymbol{y}, \boldsymbol{w})$ in $O\big(dn \log(l+m)\big)$ time. Assuming we use this construction, invoking Lemma G.3, given an optimal value of $t$, we can compute the partial argument corresponding to the values of the sequences that originate at the root infosets, which allow the optimal value to be realized for the value function. Then, we can use optimal arguments for these sequences recursively at the subtreeplexes to continue computing the optimal argument at sequences lower on the treeplex. We can do this because in the process of computing the derivative of the value function of the entire treeplex, we have also computed the derivative of the value function for each of the subtreeplexes. Thus, once we have computed an optimval value of $t$ for the value function at the top level, we can

do a top-down pass to compute the optimal values for all sequences that occur at any level in the treeplex. This is detailed in the analysis done in the proof of Lemma 14 in [17].

In order to pick the optimal value of $t$ for the value function, since $\lambda_{\mathcal{Z}}(\cdot, \boldsymbol{y}, \boldsymbol{w})$ is strictly increasing, we only have to consider two cases: $\lambda_{\mathcal{Z}}(0, \boldsymbol{y}, \boldsymbol{w}) < 0$ and $\lambda_{\mathcal{Z}}(0, \boldsymbol{y}, \boldsymbol{w}) \geq 0$. In the first case, the value function $\lambda_{\mathcal{Z}}(\cdot, \boldsymbol{y}, \boldsymbol{w})$ will be minimized when $\lambda_{\mathcal{Z}}(\cdot, \boldsymbol{y}, \boldsymbol{w})$ is equal to 0, and this can be directly computed using the standard representation (it will be necessarily 0 somewhere because it is strictly monotone). In the second case, since $\lambda_{\mathcal{Z}}(\cdot, \boldsymbol{y}, \boldsymbol{w})$ is strictly monotone and $\lambda_{\mathcal{Z}}(0, \boldsymbol{y}, \boldsymbol{w}) \geq 0$, we must have that $\lambda_{\mathcal{Z}}(\cdot, \boldsymbol{y}, \boldsymbol{w}) \geq 0$, which means that $v_{\mathcal{Z}}(\cdot, \boldsymbol{y}, \boldsymbol{w})$ is minimized at $t^* = 0$. $\qquad \square$

# H   Practical Implementation of `Smooth PTB`[+]

We have the following lemma, which shows that the stable region $\mathcal{C}_{\geq}$ admits a relatively simple formulation.

**Lemma H.1.** *The stable region*

$$\mathcal{C}_{\geq} := \mathsf{cone}(\mathcal{T}) \cap \{\boldsymbol{R} \in \mathbb{R}^{n+1} \mid \langle \boldsymbol{R}, \boldsymbol{a} \rangle \geq R_0\}$$

*can be reformulated as follows:*

$$\mathcal{C}_{\geq} = \{\alpha \boldsymbol{x} \mid \alpha \geq R_0, \boldsymbol{x} \in \mathcal{T}\}$$
$$= \{\boldsymbol{x} \in \mathbb{R}_+^{n+1} \mid x_\varnothing \geq R_0, \sum_{a \in \mathcal{A}_j} x_{ja} = x_{p_j}, \forall j \in \mathcal{J}\}.$$

*Proof.* By definition, we have

$$\mathcal{C}_{\geq} = \{\boldsymbol{R} \in \mathsf{cone}(\mathcal{T}) \mid \langle \boldsymbol{R}, \boldsymbol{a} \rangle \geq R_0\}.$$

Note that for $\boldsymbol{R} \in \mathsf{cone}(\mathcal{T})$, $\boldsymbol{R} = \alpha \boldsymbol{x}$ with $\alpha \geq 0$ and $\langle \boldsymbol{x}, \boldsymbol{a} \rangle = 1$. Therefore, for $\boldsymbol{R} \in \mathcal{C}$ we have $\langle \boldsymbol{R}, \boldsymbol{a} \rangle \geq R_0 \iff \alpha \geq R_0$. This shows that we can write

$$\mathcal{C}_{\geq} = \{\alpha \boldsymbol{x} \mid \alpha \geq R_0, \boldsymbol{x} \in \mathcal{T}\}.$$

Now let $\boldsymbol{x} \in \mathcal{C}_{\geq}$, i.e., let $\boldsymbol{x} = \alpha \hat{\boldsymbol{x}}$ with $\alpha \geq R_0$ and $\boldsymbol{x} \in \mathcal{T}$. Since $\hat{\boldsymbol{x}} \in \mathcal{T}$, we have $x_\varnothing = 1$, so that $\hat{x}_\varnothing = \alpha \geq R_0$. Additionally, we have $\hat{\boldsymbol{x}} \geq 0, \sum_{a \in \mathcal{A}_j} \hat{x}_{ja} = \hat{x}_{p_j}, \forall j \in \mathcal{J}$. Multiplying by $\alpha \geq R_0$, we obtain that $\boldsymbol{x} \geq 0$ and $\sum_{a \in \mathcal{A}_j} x_{ja} = x_{p_j}, \forall j \in \mathcal{J}$. Overall we have shown

$$\mathcal{C}_{\geq} \subseteq \{\boldsymbol{x} \in \mathbb{R}^{n+1} \mid x_\varnothing \geq R_0, \sum_{a \in \mathcal{A}_j} x_{ja} = x_{p_j}, \forall j \in \mathcal{J}, \boldsymbol{x} \geq \boldsymbol{0}\}.$$

We now consider $\boldsymbol{x} \in \{\boldsymbol{x} \in \mathbb{R}^{n+1} \mid x_\varnothing \geq R_0, \sum_{a \in \mathcal{A}_j} x_{ja} = x_{p_j}, \forall j \in \mathcal{J}, \boldsymbol{x} \geq \boldsymbol{0}\}$ with $\boldsymbol{x} \neq \boldsymbol{0}$. Then $\boldsymbol{x} = \alpha \frac{\boldsymbol{x}}{\alpha}$ with $\alpha = x_\varnothing$, so that $\alpha \geq R_0$ and

$$\sum_{a \in \mathcal{A}_j} x_{ja} = x_{p_j}, \forall j \in \mathcal{J} \iff \sum_{a \in \mathcal{A}_j} \frac{x_{ja}}{\alpha} = \frac{x_{p_j}}{\alpha}, \forall j \in \mathcal{J}.$$

Therefore

$$\{\boldsymbol{x} \in \mathbb{R}^{n+1} \mid x_\varnothing \geq R_0, \sum_{a \in \mathcal{A}_j} x_{ja} = x_{p_j}, \forall j \in \mathcal{J}, \boldsymbol{x} \geq \boldsymbol{0}\} \subseteq \mathcal{C}_{\geq}.$$

This shows that we have

$$\mathcal{C}_{\geq} = \{\boldsymbol{x} \in \mathbb{R}^{n+1} \mid x_\varnothing \geq R_0, \sum_{a \in \mathcal{A}_j} x_{ja} = x_{p_j}, \forall j \in \mathcal{J}, \boldsymbol{x} \geq \boldsymbol{0}\}.$$

$\qquad \square$

**Proposition H.2.** *For a treeplex $\mathcal{T}$ with depth $d$, number of sequences $n$, number of leaf sequences $l$, and number of infosets $m$, the complexity of computing the orthogonal projection of a point $y \in \mathbb{R}^{n+1}$ onto $\mathcal{C}_{\geq} = \{\alpha \boldsymbol{x} \mid \alpha \geq R_0, \boldsymbol{x} \in \mathcal{T}\}$ is $O(dn \log(l + m))$.*

*Proof.* The proof is the same as that for Proposition 4.4, since the derivative of the value function can be computed in $O\big(dn\log(l+m)\big)$ time. However, this time, we have an additional constraint that $t \geq R_0$. Thus instead of checking the sign of $\lambda_{\mathcal{Z}}(\cdot, \boldsymbol{y}, \boldsymbol{w})$ at $t = 0$, we check the sign at $R_0$.

If $\lambda_{\mathcal{Z}}(R_0, \boldsymbol{y}, \boldsymbol{w}) < 0$, then because $\lambda_{\mathcal{Z}}(\cdot, \boldsymbol{y}, \boldsymbol{w})$ is a strictly monotone function, the function will be $0$ for some value of $t$, and this is exactly $t^*$, which minimizes the value function with respect to $t$, when $t \geq R_0$. On the other hand, if $\lambda_{\mathcal{Z}}(R_0, \boldsymbol{y}, \boldsymbol{w}) \geq 0$, then again because the function is strictly monotone in $t$, we know that the value function must get minimized at $t^* = R_0$. Using the same argument as in the proof of Proposition 4.4, since we have computed the standard representations of the derivatives of the value functions at all of the treeplexes, we can do a top-down pass to compute the argument which leads to the optimal value of the value function.

$\square$

# I   Proof of Theorem 4.7

*Proof of Theorem 4.7.* For the sake of conciseness we write $\boldsymbol{f}_t^x = \boldsymbol{f}(\boldsymbol{x}_t, \boldsymbol{M}\boldsymbol{y}_t)$ and $\boldsymbol{f}_t^y = \boldsymbol{f}(\boldsymbol{y}_t, -\boldsymbol{M}^\top \boldsymbol{x}_t)$.

From our Proposition 4.2, we have that, for the first player,

$$\sum_{t=1}^{T} \langle \boldsymbol{x}_t - \hat{\boldsymbol{x}}, \boldsymbol{M}\boldsymbol{y}_t \rangle = \sum_{t=1}^{T} \langle \boldsymbol{R}_t - \hat{\boldsymbol{R}}, \boldsymbol{f}_t^x \rangle.$$

Now $\sum_{t=1}^{T} \langle \boldsymbol{R}_t - \hat{\boldsymbol{R}}, \boldsymbol{f}_t^x \rangle$ is the regret obtained by running Predictive OMD on $\mathcal{C}_{\geq}$ against the sequence of loss $\boldsymbol{f}_1^x, ..., \boldsymbol{f}_T^x$. From Proposition 5 in [16], we have that

$$\sum_{t=1}^{T} \langle \boldsymbol{R}_t^x - \hat{\boldsymbol{R}}^x, \boldsymbol{f}_t^x \rangle \leq \frac{\|\hat{\boldsymbol{R}}\|_2^2}{2\eta} + \eta \sum_{t=1}^{T} \|\boldsymbol{f}_t^x - \boldsymbol{f}_{t-1}^x\|_2^2 - \frac{1}{8\eta} \sum_{t=1}^{T} \|\boldsymbol{R}_{t+1}^x - \boldsymbol{R}_{t+1}^x\|_2^2.$$

Since $\hat{\boldsymbol{R}}_t \in \mathcal{C}_{\geq}$, we can use our Proposition 4.5 to show that

$$\|\boldsymbol{x}_{t+1} - \boldsymbol{x}_t\|_2^2 \leq \frac{\Omega}{R_0^2} \|\boldsymbol{R}_{t+1}^x - \boldsymbol{R}_{t+1}^x\|_2^2.$$

This shows that

$$\sum_{t=1}^{T} \langle \boldsymbol{R}_t^x - \hat{\boldsymbol{R}}^x, \boldsymbol{f}_t^x \rangle \leq \frac{\|\hat{\boldsymbol{R}}\|_2^2}{2\eta} + \eta \sum_{t=1}^{T} \|\boldsymbol{f}_t^x - \boldsymbol{f}_{t-1}^x\|_2^2 - \frac{R_0^2}{8\Omega^2\eta} \sum_{t=1}^{T} \|\boldsymbol{R}_{t+1}^x - \boldsymbol{R}_{t+1}^x\|_2^2$$

which gives, using the norm equivalence $\|\cdot\|_2 \leq \|\cdot\|_1 \leq \sqrt{n+1}\|\cdot\|_2$, the following inequality:

$$\sum_{t=1}^{T} \langle \boldsymbol{R}_t^x - \hat{\boldsymbol{R}}^x, \boldsymbol{f}_t^x \rangle \leq \frac{\|\hat{\boldsymbol{R}}\|_2^2}{2\eta} + \eta \sum_{t=1}^{T} \|\boldsymbol{f}_t^x - \boldsymbol{f}_{t-1}^x\|_1^2 - \frac{R_0^2}{8\Omega^2(n+1)\eta} \sum_{t=1}^{T} \|\boldsymbol{R}_{t+1}^x - \boldsymbol{R}_{t+1}^x\|_2^2$$

The above inequality is a RVU bound:

$$\sum_{t=1}^{T} \langle \boldsymbol{R}_t^x - \hat{\boldsymbol{R}}^x, \boldsymbol{f}_t^x \rangle \leq \alpha + \beta \sum_{t=1}^{T} \|\boldsymbol{f}_t^x - \boldsymbol{f}_{t-1}^x\|_1^2 - \gamma \sum_{t=1}^{T} \|\boldsymbol{R}_{t+1}^x - \boldsymbol{R}_{t+1}^x\|_2^2$$

with

$$\alpha = \frac{\|\hat{\boldsymbol{R}}\|_2^2}{2\eta}, \beta = \eta, \gamma = \frac{R_0^2}{8\Omega^2(n+1)\eta}. \tag{13}$$

To invoke Theorem 4 in [36], we also need the utilities of each player to be bounded by 1. This can be done can rescaling $\boldsymbol{f}_t^x = \boldsymbol{M}\boldsymbol{y}_t$ and $\boldsymbol{f}_t^y = -\boldsymbol{M}^\top \boldsymbol{x}_t$. In particular, we know that

$$\|\boldsymbol{M}\boldsymbol{y}\|_\infty \leq \|\boldsymbol{M}\|_{\ell_2, \ell_\infty} \|\boldsymbol{y}\|_2 \leq \|\boldsymbol{M}\|_{\ell_2, \ell_\infty} \cdot \hat{\Omega}$$

with $\|\boldsymbol{M}\|_{\ell_2,\ell_\infty} = \max_{i\in[n+1]}\|(A_{ij})_{j\in[m+1]}\|_2$ and $\hat{\Omega} = \max\{\max\{\|\boldsymbol{x}\|_2,\|\boldsymbol{y}\|_2\}\ \boldsymbol{x} \in \mathcal{X}, \boldsymbol{y} \in \mathcal{Y}\}$. This corresponds to multiplying $\beta$ in (13) by $\|\boldsymbol{M}\| \times \hat{\Omega}$ with $\|\boldsymbol{M}\| := \max\{\|\boldsymbol{M}\|_{\ell_2,\ell_\infty}, \|\boldsymbol{M}^\top\|_{\ell_2,\ell_\infty}\}$. To apply Theorem 4 in [36] we also need $\beta \leq \gamma$. Since we need the same condition for the second player, we take

$$\eta = R_0 \left(\sqrt{8d\hat{\Omega}^3}\|\boldsymbol{M}\|\right)^{-1}.$$

Under this condition on the stepsize, we can invoke Theorem 4 in [36] to conclude that

$$\sum_{t=1}^{T}\langle \boldsymbol{R}_t^x - \hat{\boldsymbol{R}}^x, \boldsymbol{f}_t^x\rangle + \sum_{t=1}^{T}\langle \boldsymbol{R}_t^y - \hat{\boldsymbol{R}}^y, \boldsymbol{f}_t^y\rangle \leq \frac{\|\hat{\boldsymbol{R}}^x\|_2^2 + \|\hat{\boldsymbol{R}}^y\|_2^2}{\eta}.$$

Since the duality gap is bounded by the average of the sum of the regrets of both players [19], and replacing $\eta$ by its expression, we obtain that

$$\max_{\boldsymbol{y}\in\mathcal{Y}}\langle \bar{\boldsymbol{x}}_T, \boldsymbol{M}\boldsymbol{y}\rangle - \min_{\boldsymbol{x}\in\mathcal{X}}\langle \boldsymbol{x}, \boldsymbol{M}\bar{\boldsymbol{y}}_T\rangle \leq \frac{2\hat{\Omega}^2}{\eta}\frac{1}{T}.$$

$\square$

# J   AdaGradTB⁺ and AdamTB⁺

AdaGradTB⁺.   We introduce AdaGradTB⁺ in Algorithm 6. Given matrix $\boldsymbol{A}$ and a vector $\boldsymbol{y}\in\mathbb{R}^{n+1}$, let $\mathrm{diag}(\boldsymbol{y})$ be the diagonal matrix with $\boldsymbol{y}$ on its diagonal and $\Pi_{\mathcal{C}}^{A}(\boldsymbol{y}) = \arg\min_{\boldsymbol{x}\in\mathcal{C}}\langle \boldsymbol{x}-\boldsymbol{y}, \boldsymbol{A}(\boldsymbol{x}-\boldsymbol{y})\rangle$. We first show that AdaGradTB⁺ is a regret minimizer.

**Proposition J.1.** *Let* $\boldsymbol{x}_1,...,\boldsymbol{x}_T$ *be computed by* AdaGradTB⁺. *For* $\eta = \frac{\max_{t\leq T}(\|\boldsymbol{R}_t\|_2+\Omega)^2}{\sqrt{2}}$, *we have*

$\max_{\hat{\boldsymbol{x}}\in\mathcal{T}}\sum_{t=1}^{T}\langle \boldsymbol{x}_t - \hat{\boldsymbol{x}}, \boldsymbol{\ell}_t\rangle \leq 2\eta\sum_{i=1}^{d}\sqrt{\sum_{t=1}^{T}\left(\boldsymbol{f}_t(\boldsymbol{x}_t,\boldsymbol{\ell}_t)\right)_i^2}$.

We omit the proof of Proposition J.1 for conciseness; it follows from the regret guarantees of AdaGrad (Theorem 5 in [12]) and Proposition 3.1. We conclude that combining AdaGradTB⁺ with the self-play framework ensures a $O(1/\sqrt{T})$ convergence rate.

---

**Algorithm 6** AdaGradTB⁺

---

1: **Input**: $\eta, \delta > 0$
2: **Initialization**: $\boldsymbol{R}_1 = \boldsymbol{s}_0 = \boldsymbol{g}_0 = \boldsymbol{0} \in \mathbb{R}^{n+1}$
3: **for** $t = 1,\ldots,T$ **do**
4:     $\boldsymbol{x}_t = \boldsymbol{R}_t/\langle \boldsymbol{R}_t, \boldsymbol{a}\rangle$
5:     Observe the loss vector $\boldsymbol{\ell}_t \in \mathbb{R}^{n+1}$
6:     $\boldsymbol{s}_t = \boldsymbol{s}_{t-1} + \boldsymbol{f}(\boldsymbol{x}_t,\boldsymbol{\ell}_t) \odot \boldsymbol{f}(\boldsymbol{x}_t,\boldsymbol{\ell}_t)$
7:     $\boldsymbol{H}_t = \mathrm{diag}\left(\sqrt{\boldsymbol{s}_t} + \epsilon\boldsymbol{1}\right)$
8:     $\boldsymbol{R}_{t+1} \in \Pi_{\mathcal{C}}^{\boldsymbol{H}_t}\left(\boldsymbol{R}_t - \eta\boldsymbol{H}_t^{-1}\boldsymbol{f}(\boldsymbol{x}_t,\boldsymbol{\ell}_t)\right)$

---

AdamTB⁺.   We present AdamTB⁺, our instantiation of Algorithm 1 inspired from the adaptive algorithm Adam [25] in Algorithm 7. Since Adam is not necessarily a regret minimizer [35], there are no regret guarantees for AdamTB⁺. We choose to consider this algorithm for the sake of completeness, since Adam is widely used in other settings.

# K   Details on Numerical Experiments

## K.1   Additional Algorithms

**Single-call Predictive Online Mirror Descent** (SC-POMD).   We present SC-POMD in Algorithm 8. This algorithm runs a variant of predictive online mirror descent with only one orthogonal projection

**Algorithm 7** AdamTB⁺

1: **Input**: $\eta, \delta > 0, \beta_1, \beta_2 \in [0, 1]$
2: **Initialization**: $\boldsymbol{R}_1 = \boldsymbol{0} \in \mathbb{R}^{n+1}, \boldsymbol{s}_0 = \boldsymbol{0} \in \mathbb{R}^{n+1}, \boldsymbol{g}_0 = \boldsymbol{0} \in \mathbb{R}^{n+1}$
3: **for** $t = 1, \ldots, T$ **do**
4:     $\boldsymbol{x}_t = \boldsymbol{R}_t / \langle \boldsymbol{R}_t, \boldsymbol{a} \rangle$
5:     Observe the loss vector $\boldsymbol{\ell}_t \in \mathbb{R}^{n+1}$
6:     $\boldsymbol{s}_t = \beta_2 \boldsymbol{s}_{t-1} + (1 - \beta_2) \boldsymbol{f}(\boldsymbol{x}_t, \boldsymbol{\ell}_t) \odot \boldsymbol{f}(\boldsymbol{x}_t, \boldsymbol{\ell}_t)$
7:     $\hat{\boldsymbol{s}}_t = \boldsymbol{s}_t / (1 - \beta_2^t)$
8:     $\boldsymbol{g}_t = \beta_1 \boldsymbol{g}_{t-1} + (1 - \beta_1) \boldsymbol{f}(\boldsymbol{x}_t, \boldsymbol{\ell}_t)$
9:     $\hat{\boldsymbol{g}}_t = \boldsymbol{g}_t / (1 - \beta_1^t)$
10:    $\boldsymbol{H}_t = \text{diag}\left(\sqrt{\hat{\boldsymbol{s}}_t} + \epsilon \boldsymbol{1}\right)$
11:    $\boldsymbol{R}_{t+1} \in \Pi_{\mathcal{C}}^{\boldsymbol{H}_t}\left(\boldsymbol{R}_t - \eta \boldsymbol{H}_t^{-1} \hat{\boldsymbol{g}}_t\right)$

at every iteration [24]. The pseudocode from Algorithm 8 corresponds to choosing the squared $\ell_2$-norm as a distance generating function - in principle, other distance generating functions are possible, e.g. dilated entropy [15]. Combined with the self-play framework, SC-POMD ensures that the average of the visited iterates converges to a Nash equilibrium at a rate of $O(1/T)$, similar to the variant of predictive online mirror descent with two orthogonal projections at every iteration [15].

**Algorithm 8** Single-call predictive online mirror descent (SC-POMD)

1: **Input**: $\eta > 0,$
2: **Initialization**: $\boldsymbol{x}_0 = \boldsymbol{\ell}_0 = \boldsymbol{\ell}_{-1} = \boldsymbol{0} \in \mathbb{R}^{n+1}$
3: **for** $t = 1, \ldots, T$ **do**
4:     $\boldsymbol{x}_t = \Pi_{\mathcal{T}}\left(\boldsymbol{x}_{t-1} - \eta\left(2\boldsymbol{\ell}_{t-1} - \boldsymbol{\ell}_{t-2}\right)\right)$
5:     Observe the loss vector $\boldsymbol{\ell}_t \in \mathbb{R}^{n+1}$

## K.2   Algorithm Implementation Details

All algorithms are initialized using the uniform strategy (placing equal probability on each action at each decision point). For algorithms that are not stepsize invariant (Smooth PTB⁺ and SC-POMD), we try stepsizes in $\eta \in \{0.05, 0.1, 0.5, 1, 2, 5\}$ and we present the performance with the best stepsize. For Smooth PTB⁺, we use $R_0 = 0.1$. For both AdaGradTB⁺ and AdamTB⁺, we use $\delta = 1 \times 10^{-6}$, and for AdamTB⁺ we use $\beta_1 = 0.9$ and $\beta_2 = 0.999$.

## K.3   Comparing the Performance of our Algorithms

In Figure 4 we compare the performance of TB⁺, PTB⁺, Smooth PTB⁺, AdaGradTB⁺ and AdamTB⁺.

It can be seen that PTB⁺ and Smooth PTB⁺ perform similarly, both when using quadratic averaging and when using the last iterate, and they generally outperform the other algorithms. In Kuhn, Liar's Dice, and Battleship, the last iterate seems to perform quite well, whereas in Leduc and Goofspiel, the quadratic averaging scheme works better. AdamTB⁺ seems to not converge in any of the games, which is not surprising, because it does not have theoretical guarantees for convergence.

## K.4   Individual Performance

In Figure 5fig:scpomd, we compare the individual performance of TB⁺, PTB⁺, Smooth PTB⁺, AdaGradTB⁺, AdamTB⁺, CFR⁺, PCFR⁺ and SC-POMD with different weighting schemes, with and without alternation. We also show the performance of the last iterate. The goal is to choose the most favorable framework for each algorithms, in order to have a fair comparison. We find that all algorithms benefit from using alternation. CFR⁺ enjoys stronger performance using linear weights, whereas PTB⁺, PCFR⁺ and SC-POMD have stronger performances with quadratic weights. For this reason this is the setup that we present for comparing the performance of these algorithms in our main body (Figure 2).

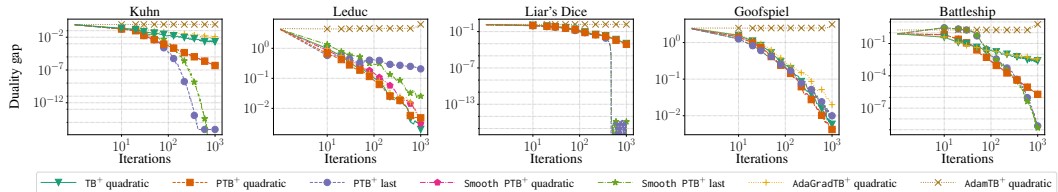

Figure 4: Convergence to Nash equilibrium as a function of number of iterations for TB$^+$ with quadratic averaging, PTB$^+$ with quadratic averaging and last iterate, and Smooth PTB$^+$ with quadratic averaging and last iterate. Every algorithm is using alternation.

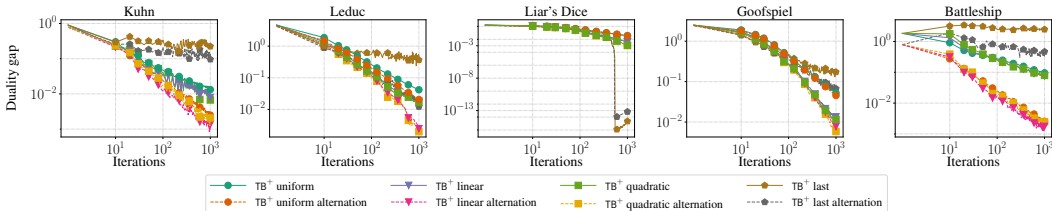

Figure 5: Convergence to Nash equilibrium as a function of number of iterations using uniform, linear, and quadratic averaging, as well as the last iterate, with and without alternation for TB$^+$.


Figure 6: Convergence to Nash equilibrium as a function of number of iterations using uniform, linear, and quadratic averaging, as well as the last iterate, with and without alternation for PTB⁺.

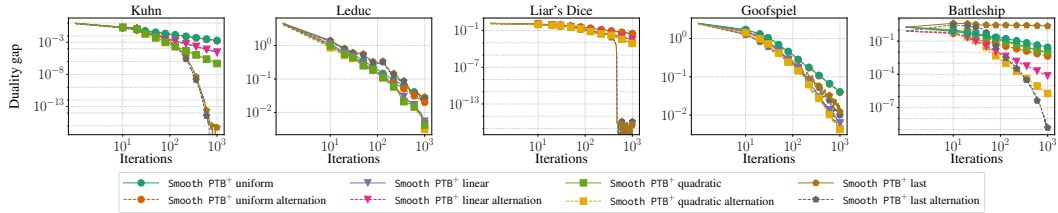

Figure 7: Convergence to Nash equilibrium as a function of number of iterations using uniform, linear, and quadratic averaging, as well as the last iterate, with and without alternation for Smooth PTB⁺.

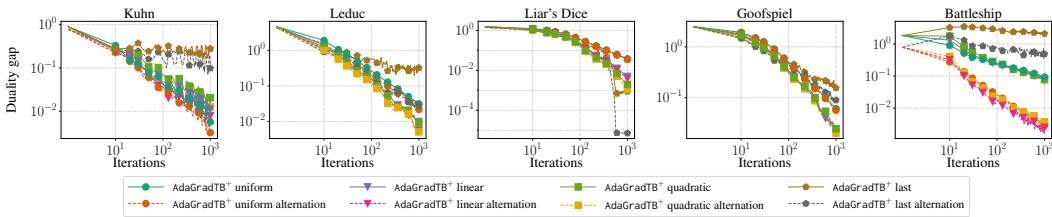

Figure 8: Convergence to Nash equilibrium as a function of number of iterations using uniform, linear, and quadratic averaging, as well as the last iterate, with and without alternation for AdaGradTB⁺.

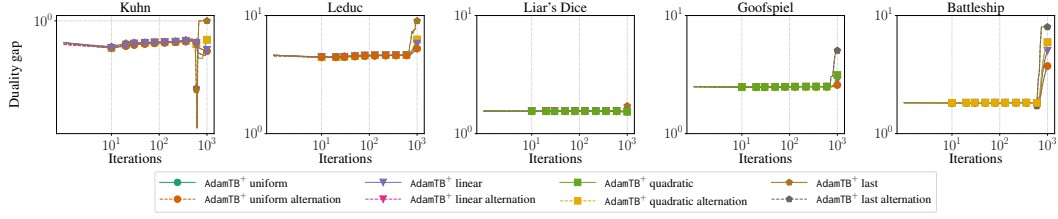

Figure 9: Convergence to Nash equilibrium as a function of number of iterations using uniform, linear, and quadratic averaging, as well as the last iterate, with and without alternation for AdaGradTB⁺.

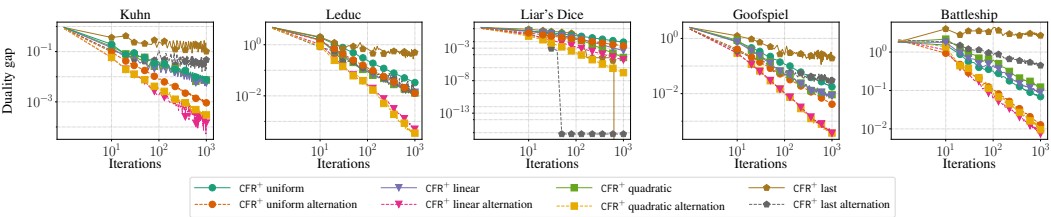

Figure 10: Convergence to Nash equilibrium as a function of number of iterations using uniform, linear, and quadratic averaging, as well as the last iterate, with and without alternation for CFR⁺.

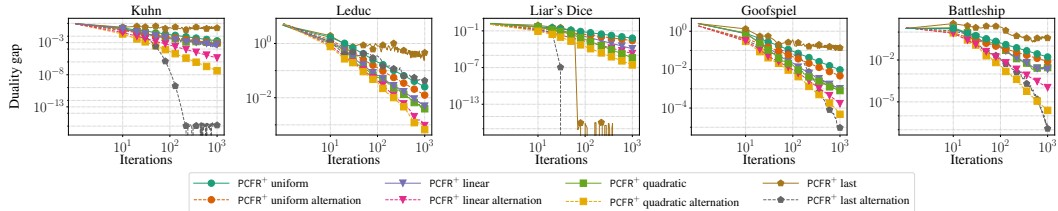

Figure 11: Convergence to Nash equilibrium as a function of number of iterations using uniform, linear, and quadratic averaging, as well as the last iterate, with and without alternation for PCFR⁺.

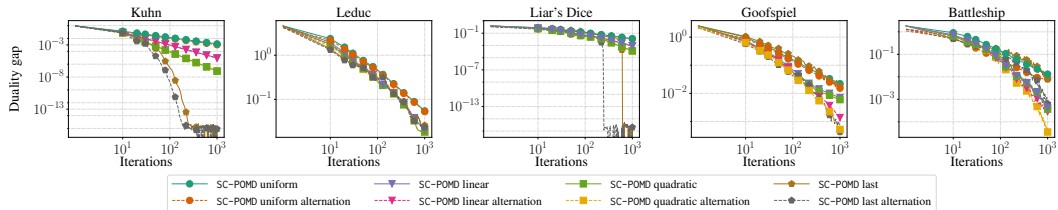

Figure 12: Convergence to Nash equilibrium as a function of number of iterations using uniform, linear, and quadratic averaging, as well as the last iterate, with and without alternation for SC-POMD.

- The authors should reflect on the scope of the claims made, e.g., if the approach was only tested on a few datasets or with a few runs. In general, empirical results often depend on implicit assumptions, which should be articulated.
- The authors should reflect on the factors that influence the performance of the approach. For example, a facial recognition algorithm may perform poorly when image resolution is low or images are taken in low lighting. Or a speech-to-text system might not be used reliably to provide closed captions for online lectures because it fails to handle technical jargon.
- The authors should discuss the computational efficiency of the proposed algorithms and how they scale with dataset size.
- If applicable, the authors should discuss possible limitations of their approach to address problems of privacy and fairness.
- While the authors might fear that complete honesty about limitations might be used by reviewers as grounds for rejection, a worse outcome might be that reviewers discover limitations that aren't acknowledged in the paper. The authors should use their best judgment and recognize that individual actions in favor of transparency play an important role in developing norms that preserve the integrity of the community. Reviewers will be specifically instructed to not penalize honesty concerning limitations.

3. **Theory Assumptions and Proofs**

Question: For each theoretical result, does the paper provide the full set of assumptions and a complete (and correct) proof?

Answer: [Yes]

Justification: This is done in the main body and in the appendix.

Guidelines:

- The answer NA means that the paper does not include theoretical results.
- All the theorems, formulas, and proofs in the paper should be numbered and cross-referenced.
- All assumptions should be clearly stated or referenced in the statement of any theorems.
- The proofs can either appear in the main paper or the supplemental material, but if they appear in the supplemental material, the authors are encouraged to provide a short proof sketch to provide intuition.
- Inversely, any informal proof provided in the core of the paper should be complemented by formal proofs provided in appendix or supplemental material.

- Theorems and Lemmas that the proof relies upon should be properly referenced.

4. **Experimental Result Reproducibility**

   Question: Does the paper fully disclose all the information needed to reproduce the main experimental results of the paper to the extent that it affects the main claims and/or conclusions of the paper (regardless of whether the code and data are provided or not)?

   Answer: [Yes]

   Justification: We do so in the main body and in the appendices (Appendix H, Appendix K).

   Guidelines:

   - The answer NA means that the paper does not include experiments.
   - If the paper includes experiments, a No answer to this question will not be perceived well by the reviewers: Making the paper reproducible is important, regardless of whether the code and data are provided or not.
   - If the contribution is a dataset and/or model, the authors should describe the steps taken to make their results reproducible or verifiable.
   - Depending on the contribution, reproducibility can be accomplished in various ways. For example, if the contribution is a novel architecture, describing the architecture fully might suffice, or if the contribution is a specific model and empirical evaluation, it may be necessary to either make it possible for others to replicate the model with the same dataset, or provide access to the model. In general. releasing code and data is often one good way to accomplish this, but reproducibility can also be provided via detailed instructions for how to replicate the results, access to a hosted model (e.g., in the case of a large language model), releasing of a model checkpoint, or other means that are appropriate to the research performed.
   - While NeurIPS does not require releasing code, the conference does require all submissions to provide some reasonable avenue for reproducibility, which may depend on the nature of the contribution. For example
     (a) If the contribution is primarily a new algorithm, the paper should make it clear how to reproduce that algorithm.
     (b) If the contribution is primarily a new model architecture, the paper should describe the architecture clearly and fully.
     (c) If the contribution is a new model (e.g., a large language model), then there should either be a way to access this model for reproducing the results or a way to reproduce the model (e.g., with an open-source dataset or instructions for how to construct the dataset).
     (d) We recognize that reproducibility may be tricky in some cases, in which case authors are welcome to describe the particular way they provide for reproducibility. In the case of closed-source models, it may be that access to the model is limited in some way (e.g., to registered users), but it should be possible for other researchers to have some path to reproducing or verifying the results.

5. **Open access to data and code**

   Question: Does the paper provide open access to the data and code, with sufficient instructions to faithfully reproduce the main experimental results, as described in supplemental material?

   Answer: [Yes]

   Justification: We plan to do so after the revision process.

   Guidelines:

   - The answer NA means that paper does not include experiments requiring code.
   - Please see the NeurIPS code and data submission guidelines (`https://nips.cc/public/guides/CodeSubmissionPolicy`) for more details.
   - While we encourage the release of code and data, we understand that this might not be possible, so "No" is an acceptable answer. Papers cannot be rejected simply for not including code, unless this is central to the contribution (e.g., for a new open-source benchmark).

- The instructions should contain the exact command and environment needed to run to reproduce the results. See the NeurIPS code and data submission guidelines (`https://nips.cc/public/guides/CodeSubmissionPolicy`) for more details.
- The authors should provide instructions on data access and preparation, including how to access the raw data, preprocessed data, intermediate data, and generated data, etc.
- The authors should provide scripts to reproduce all experimental results for the new proposed method and baselines. If only a subset of experiments are reproducible, they should state which ones are omitted from the script and why.
- At submission time, to preserve anonymity, the authors should release anonymized versions (if applicable).
- Providing as much information as possible in supplemental material (appended to the paper) is recommended, but including URLs to data and code is permitted.

6. **Experimental Setting/Details**

   Question: Does the paper specify all the training and test details (e.g., data splits, hyper-parameters, how they were chosen, type of optimizer, etc.) necessary to understand the results?

   Answer: [Yes]

   Justification: We do so in the main body and in the appendices.

   Guidelines:

   - The answer NA means that the paper does not include experiments.
   - The experimental setting should be presented in the core of the paper to a level of detail that is necessary to appreciate the results and make sense of them.
   - The full details can be provided either with the code, in appendix, or as supplemental material.

7. **Experiment Statistical Significance**

   Question: Does the paper report error bars suitably and correctly defined or other appropriate information about the statistical significance of the experiments?

   Answer: [NA]

   Justification: No random experiments.

   Guidelines:

   - The answer NA means that the paper does not include experiments.
   - The authors should answer "Yes" if the results are accompanied by error bars, confidence intervals, or statistical significance tests, at least for the experiments that support the main claims of the paper.
   - The factors of variability that the error bars are capturing should be clearly stated (for example, train/test split, initialization, random drawing of some parameter, or overall run with given experimental conditions).
   - The method for calculating the error bars should be explained (closed form formula, call to a library function, bootstrap, etc.)
   - The assumptions made should be given (e.g., Normally distributed errors).
   - It should be clear whether the error bar is the standard deviation or the standard error of the mean.
   - It is OK to report 1-sigma error bars, but one should state it. The authors should preferably report a 2-sigma error bar than state that they have a 96% CI, if the hypothesis of Normality of errors is not verified.
   - For asymmetric distributions, the authors should be careful not to show in tables or figures symmetric error bars that would yield results that are out of range (e.g. negative error rates).
   - If error bars are reported in tables or plots, The authors should explain in the text how they were calculated and reference the corresponding figures or tables in the text.

8. **Experiments Compute Resources**

Question: For each experiment, does the paper provide sufficient information on the computer resources (type of compute workers, memory, time of execution) needed to reproduce the experiments?

Answer: [Yes]

Justification: We do so in the main body and in the appendices.

Guidelines:

- The answer NA means that the paper does not include experiments.
- The paper should indicate the type of compute workers CPU or GPU, internal cluster, or cloud provider, including relevant memory and storage.
- The paper should provide the amount of compute required for each of the individual experimental runs as well as estimate the total compute.
- The paper should disclose whether the full research project required more compute than the experiments reported in the paper (e.g., preliminary or failed experiments that didn't make it into the paper).

9. **Code Of Ethics**

Question: Does the research conducted in the paper conform, in every respect, with the NeurIPS Code of Ethics https://neurips.cc/public/EthicsGuidelines?

Answer: [Yes]

Justification: The research conducted in the paper conform with the NeurIPS Code of Ethics.

Guidelines:

- The answer NA means that the authors have not reviewed the NeurIPS Code of Ethics.
- If the authors answer No, they should explain the special circumstances that require a deviation from the Code of Ethics.
- The authors should make sure to preserve anonymity (e.g., if there is a special consideration due to laws or regulations in their jurisdiction).

10. **Broader Impacts**

Question: Does the paper discuss both potential positive societal impacts and negative societal impacts of the work performed?

Answer: [NA]

Justification: This research focuses on foundational questions. There is no direct societal impacts of the work performed.

Guidelines:

- The answer NA means that there is no societal impact of the work performed.
- If the authors answer NA or No, they should explain why their work has no societal impact or why the paper does not address societal impact.
- Examples of negative societal impacts include potential malicious or unintended uses (e.g., disinformation, generating fake profiles, surveillance), fairness considerations (e.g., deployment of technologies that could make decisions that unfairly impact specific groups), privacy considerations, and security considerations.
- The conference expects that many papers will be foundational research and not tied to particular applications, let alone deployments. However, if there is a direct path to any negative applications, the authors should point it out. For example, it is legitimate to point out that an improvement in the quality of generative models could be used to generate deepfakes for disinformation. On the other hand, it is not needed to point out that a generic algorithm for optimizing neural networks could enable people to train models that generate Deepfakes faster.
- The authors should consider possible harms that could arise when the technology is being used as intended and functioning correctly, harms that could arise when the technology is being used as intended but gives incorrect results, and harms following from (intentional or unintentional) misuse of the technology.

- If there are negative societal impacts, the authors could also discuss possible mitigation strategies (e.g., gated release of models, providing defenses in addition to attacks, mechanisms for monitoring misuse, mechanisms to monitor how a system learns from feedback over time, improving the efficiency and accessibility of ML).

11. **Safeguards**

Question: Does the paper describe safeguards that have been put in place for responsible release of data or models that have a high risk for misuse (e.g., pretrained language models, image generators, or scraped datasets)?

Answer: [NA]

Justification: Not applicable to our paper. There is no release of data.

Guidelines:

- The answer NA means that the paper poses no such risks.
- Released models that have a high risk for misuse or dual-use should be released with necessary safeguards to allow for controlled use of the model, for example by requiring that users adhere to usage guidelines or restrictions to access the model or implementing safety filters.
- Datasets that have been scraped from the Internet could pose safety risks. The authors should describe how they avoided releasing unsafe images.
- We recognize that providing effective safeguards is challenging, and many papers do not require this, but we encourage authors to take this into account and make a best faith effort.

12. **Licenses for existing assets**

Question: Are the creators or original owners of assets (e.g., code, data, models), used in the paper, properly credited and are the license and terms of use explicitly mentioned and properly respected?

Answer: [Yes]

Justification: We provide citations to the packages we use.

Guidelines:

- The answer NA means that the paper does not use existing assets.
- The authors should cite the original paper that produced the code package or dataset.
- The authors should state which version of the asset is used and, if possible, include a URL.
- The name of the license (e.g., CC-BY 4.0) should be included for each asset.
- For scraped data from a particular source (e.g., website), the copyright and terms of service of that source should be provided.
- If assets are released, the license, copyright information, and terms of use in the package should be provided. For popular datasets, paperswithcode.com/datasets has curated licenses for some datasets. Their licensing guide can help determine the license of a dataset.
- For existing datasets that are re-packaged, both the original license and the license of the derived asset (if it has changed) should be provided.
- If this information is not available online, the authors are encouraged to reach out to the asset's creators.

13. **New Assets**

Question: Are new assets introduced in the paper well documented and is the documentation provided alongside the assets?

Answer: [NA]

Justification: Not applicable to our paper.

Guidelines:

- The answer NA means that the paper does not release new assets.

- Researchers should communicate the details of the dataset/code/model as part of their submissions via structured templates. This includes details about training, license, limitations, etc.
- The paper should discuss whether and how consent was obtained from people whose asset is used.
- At submission time, remember to anonymize your assets (if applicable). You can either create an anonymized URL or include an anonymized zip file.

14. **Crowdsourcing and Research with Human Subjects**

   Question: For crowdsourcing experiments and research with human subjects, does the paper include the full text of instructions given to participants and screenshots, if applicable, as well as details about compensation (if any)?

   Answer: [NA]

   Justification: Not applicable to our paper.

   Guidelines:
   - The answer NA means that the paper does not involve crowdsourcing nor research with human subjects.
   - Including this information in the supplemental material is fine, but if the main contribution of the paper involves human subjects, then as much detail as possible should be included in the main paper.
   - According to the NeurIPS Code of Ethics, workers involved in data collection, curation, or other labor should be paid at least the minimum wage in the country of the data collector.

15. **Institutional Review Board (IRB) Approvals or Equivalent for Research with Human Subjects**

   Question: Does the paper describe potential risks incurred by study participants, whether such risks were disclosed to the subjects, and whether Institutional Review Board (IRB) approvals (or an equivalent approval/review based on the requirements of your country or institution) were obtained?

   Answer: [NA]

   Justification: Not applicable to our paper.

   Guidelines:
   - The answer NA means that the paper does not involve crowdsourcing nor research with human subjects.
   - Depending on the country in which research is conducted, IRB approval (or equivalent) may be required for any human subjects research. If you obtained IRB approval, you should clearly state this in the paper.
   - We recognize that the procedures for this may vary significantly between institutions and locations, and we expect authors to adhere to the NeurIPS Code of Ethics and the guidelines for their institution.
   - For initial submissions, do not include any information that would break anonymity (if applicable), such as the institution conducting the review.

