# OpenReview forum: "Extensive-Form Game Solving via Blackwell Approachability on Treeplexes"
_NeurIPS.cc/2024/Conference — NeurIPS 2024 spotlight_

### Official Review · Reviewer_qS3q · 2024-07-03

**Soundness:** 3
**Presentation:** 3
**Contribution:** 3
**Rating:** 6
**Confidence:** 4

**Summary:**

This paper introduces a method, predictive treeplex Blackwell, for using Blackwell approachability directly on the treeplex in order to perform regret minimization in extensive-form strategy sets. They show that their algorithm achieves $O(\sqrt{T})$ regret (where $O$ hides polynomial factors in the game size), and a smoothed version of their algorithm enjoys $O(1/T)$ convergence toward Nash equilibrium when used by both players in zero-sum game.

**Strengths:**

The method is conceptually very interesting, and shows that Blackwell-based stepsize-invariant regret minimizers are not solely restricted to the simplex. The experiments are comprehensive and illuminate clearly the authors' message about the role of infoset-level stepsize invariance. Thus, despite some minor issues listed below, I am generally in favor of acceptance.

**Weaknesses:**

My most significant issue is simple: although the method is conceptually interesting, it is unclear if it carries any advantages over CFR (see also questions below re. clairvoyant CFR). For example, in theory, by Corollary 4.3 it seems that the convergence rate is something like $O(d^{5/2}/\sqrt{T})$ (since $\hat\Omega \le \sqrt{d}; \lVert \boldsymbol M \rVert_2 \le d$ -- maybe tighter analysis is possible but my point would stand regardless), compared to the $O(d/\sqrt{T})$ achieved by CFR-based methods. Similarly, the method seems consistently outperformed or matched by PCFR+ in basically every game, and the per-iteration complexity is inferior by a logarithmic factor.

Minor notes:
1. In Proposition 3.1, the regret of Algorithm 1 should be expressed in terms of the regret of the regret minimizer on $\mathcal C$. Also, technically, $\text{cone}(\mathcal T)$ is an infinite set, so by "regret of  the regret minimizer on $\mathcal C$" I really mean "regret of the regret minimizer on $\mathcal C$ against vectors $\boldsymbol{\hat x} \in \mathcal T$". The proper regret bound holds for OMD/FTRL with gradient dynamics, but these details, I think, should be spelled out. See e.g. Proposition 2 and the discussion afterward in [12].

**Questions:**

1. The paper claims that Smooth PTB+ is the first Blackwell-based $O(1/T)$-converging algorithm for EFGs. But doesn't the algorithm it is based on, namely the Clairvoyant CFR+ algorithm of [14], also achieve the same convergence rate? (see Appendix J of [14]).
1. It is interesting to me that RM+ does not coincide with TB+ when the domain is a simplex (Appendix E). What happens experimentally if you use (P)TB+ instead of (P)RM+ at every information set in CFR?
1. Is it possible to extend the fundamental ideas of this paper beyond treeplexes to other convex sets, to obtain more stepsize-invariant regret minimizers?

**Limitations:**

yes

---

> ### Author Rebuttal · Authors · 2024-08-01
>
> We thank for your time reviewing the paper and for your encouraging comments. We answer your questions below.
>
> ### Response to questions.
>
> * *My most significant issue is simple: although the method is conceptually interesting, it is unclear if it carries any advantages over CFR (see also questions below re. clairvoyant CFR).*
>
> Thanks for mentioning this. It is correct that our proposed algorithms may not outperform CFR on every front:
>
> * In terms of theoretical properties, Smooth PTB+ achieves a faster $O(1/T)$ convergence rate, superior to the $O(1/\sqrt{T})$ rate for CFR-based algorithms. However, Smooth PTB+ loses stepsize invariance.
> * In terms of empirical convergence/practical performance, PTB+ is our best algorithm but it is outperformed by PCFR+ for solving the game instances that we have tried.
>
> We would like to emphasize that one of our main objectives in this paper is to provide a novel understanding of CFR-based algorithms; beating PCFR+ is not our main target, and this has been observed as very difficult repeatedly in prior work. In light of this, we view our main contribution as providing the first coherent hypothesis for the strong practical performances of PCFR+. To do so, we first introduce the first *theoretical* distinction between infoset stepsize invariance and treeplex stepsize invariance. We then explore numerically the impact of these properties. We first run a set of extensive numerical experiments to notice that among all the algorithms studied in our framework (PTB+, Smooth PTB+, AdagradTB+, AdamTB+) the best one is PTB+, the only (treeplex) stepsize invariance one. We then run additional experiments to highlight that PCFR+ outperforms PTB+, emphasizing the *practical* distinction between infoset vs. treeplex stepsize invariance.
>
> **Questions**:
>
> * *The paper claims that Smooth PTB+ is the first Blackwell-based O(1/T)-converging algorithm for EFGs. But doesn't the algorithm it is based on, namely the Clairvoyant CFR+ algorithm of [14], also achieve the same convergence rate? (see Appendix J of [14]).*
>
> Thanks for mentioning this point. The Clairvoyant CFR (CCFR) algorithm from [14] is also based on Blackwell approachability, but on Blackwell approachability *over simplexes*, since it is based on the CFR decomposition, which enables the decision-maker to run independent regret minimizers at each information set, where the decision set is a simplex. It is true that [14] achieves $O(1/T)$ convergence for games played on simplexes with a similar type of "Blackwell-based" algorithm in self-play. However, it is not true that Clairvoyant CFR achieves a $O(1/T)$ rate on EFGs. Instead, it achieves a $O(\log T / T)$ rate. This $\log T$ dependence occurs because each outer iteration of CCFR itself must solve a fixed-point problem, and thus the algorithm does not even fall under the category of "self-play via regret minimization" algorithms. Practically speaking, this is not very desirable. In contrast, we achieve $O(1/T)$ in the sense of e.g. optimistic FTRL or OMD, where we only require simple repeated self-play. We will make this more precise in our revision.
>
> * *It is interesting to me that RM+ does not coincide with TB+ when the domain is a simplex (Appendix E). What happens experimentally if you use (P)TB+ instead of (P)RM+ at every information set in CFR?*
>
> TB+ implemented on the simplex is similar to the CBA+ algorithm from [17] (though not quite the same due to the lifting procedure in [17] being different). The authors in [17] provide numerical experiments combining the CFR decomposition with CBA+ as a regret minimizer at the infoset level, see Figure 3 and Figure 4 in [17], which show that this algorithm may outperform CFR+ in terms of the duality gap *after $T$ iterations*, but is outperformed by CFR+ in terms of the duality gap *after the same computation time*. We emphasize that [17] do not study the treeplex/EFG settings, which is the main focus of our work, and for which we can prove faster convergence rates and distinguish between interesting stepsize invariance properties. We will add a remark on this in our section on numerical experiments.
>
> * *Is it possible to extend the fundamental ideas of this paper beyond treeplexes to other convex sets, to obtain more stepsize-invariant regret minimizers?*
>
> Our ideas can be extended directly to any convex *compact* decision sets, as can be verified by inspecting the proof of Proposition 3.1. We only need the decision set $X$ to be convex, compact, and such that we can find a vector $a$ such that $\langle a,x\rangle=1$ for any $x \in X$. To ensure this last condition, we can augment the decision set $X$ with an extra dimension, i.e. we can consider the $X’ = ${$1$} $\times X$ and $a=(1,\boldsymbol{0})$. In this sense, our framework can be extended to work with any convex compact sets, and the regret bounds shown in our main propositions and theorems still hold with $\Omega$ the maximum $\ell_2$ norm of elements in $X$.
> Thanks for mentioning this interesting point. We will discuss it at the end of our revised paper.
>
> [13] G. Farina, J. Grand-Clément, C. Kroer, C.-W. Lee, and H. Luo. Regret matching+: (in)stability
> and fast convergence in games. In Advances in Neural Information Processing Systems, 2023.
>
> [17] J. Grand-Clement and C. Kroer. Solving optimization problems with Blackwell approachability. Mathematics of Operations Research, 2023.]
>
>
> ### Conclusion.
>
> We thank you again for your interesting comments, which will lead to an improved paper. Please let us know if you have any questions.

---

> > ### Comment · Reviewer_qS3q · 2024-08-08
> >
> > Thank you. My opinion of the paper was and remains positive, and I will keep my score.

---

### Official Review · Reviewer_PEWS · 2024-07-11

**Soundness:** 4
**Presentation:** 4
**Contribution:** 3
**Rating:** 7
**Confidence:** 4

**Summary:**

This paper studies a Blackwell's approachability method for solving extensive-form game. Rather than applying it at each infoset as in the CFR approach, it applies it globally, which allows for a $\mathcal{O}(1/T)$ convergence rate with a predictive version.

**Strengths:**

This paper is well written and well presented.

The results are sound, and the authors are the first to obtain a $\mathcal{O}(1/T)$ regret .

It is surprising that such an approach has not been published before.

**Weaknesses:**

The fact that CFR+ obtains good practical results because no learning tuning is needed is more or less already known.

The experiments are a bit hard to read.

**Questions:**

The conclusion mentions that the algorithm gets worse practical results than the CFR approaches, despite the stepsize invariance. This implies that the invariance may be necessary at the infoset level. Do you think it is possible to change your approach to allow for such infoset invariance?

**Limitations:**

The authors mentioned the limit of their approach (mainly the invariance mentioned above). There is no potential negative societal impact behind their work.

---

> ### Author Rebuttal · Authors · 2024-08-01
>
> Thanks for your positive reviews. We answer your questions below, quoting them in italics and answering in regular font. We remain available during the rebuttal period if you have any other questions/comments.
>
> * *The experiments are a bit hard to read.*
>
> In the camera-ready version, we will use our additional page to describe in more detail our numerical experiments.
>
> * *The conclusion mentions that the algorithm gets worse practical results than the CFR approaches, despite the stepsize invariance. This implies that the invariance may be necessary at the infoset level. Do you think it is possible to change your approach to allow for such infoset invariance?*
>
> Thanks for this interesting question. We hypothesize that obtaining an infoset stepsize invariant algorithm may require a different reduction from the one we propose in Proposition 3.1, which relates the regret in the simplex $\mathcal{T}$ to the regret in the cone $cone(\mathcal{T})$. To obtain an infoset stepsize invariant algorithm, it may necessary to relate the regret in $\mathcal{T}$ to the regret in another conic set different from $cone(\mathcal{T})$, but we were not able to provide a complete answer here. We will include this question in a discussion section at the end of our revised version.

---

### Official Review · Reviewer_n7s4 · 2024-07-14

**Soundness:** 4
**Presentation:** 3
**Contribution:** 2
**Rating:** 4
**Confidence:** 4

**Summary:**

This paper designs a new algorithm for computing minimax equilibria in two-player zero-sum extensive form games. It is well-known that one way to do this is to have both players run no-regret algorithms against each other and take the time-average of their strategies. The main innovation of this paper is designing a new class of no-regret algorithms for extensive form games via a reduction from Blackwell approachability.

In particular, minimizing regret in a zero-sum extensive form game is equivalent to minnimizing external regret in an online linear optimization problem over a polytope called the treeplex. The authors show that if you cast this OLO problem as a Blackwell approachability problem, and then perform an approachability to OLO reduction (along the lines of Abernethy et al.), you can reduce the original OLO problem of minimizing external regret over the treeplex to minimizing external regret over the conical hull of the treeplex.

If you use predictive online mirror descent (with Euclidean distance) to solve the resulting OLO problem, you get an algorithm for the original problem the authors call Predictive Treeplex Blackwell+ (PTB+). This algorithm is stepsize independent and has O(1/sqrt(T)) convergence to equilibria. If you “smooth” this algorithm by projecting onto a truncation of the cone, you get a different algorithm the authors call Smooth PTB+, which the authors show has O(1/T) convergence to equilibria (but is no longer stepsize invariant).

The authors implement these algorithms (and some other transformations of existing algorithms, e.g. Adam and AdaGrad) and compare them experimentally to a range of existing algorithms. They find that PTB+ performs the best out of these new algorithms, but still is significantly worse than the state-of-the-art PCFR+ (predictive counterfactual regret minimization) on some games. The authors hypothesize this is due to the fact that PCFR+ has an even stronger form of stepsize invariance than PTB+.

**Strengths:**

Extensive-form game solving is one of the big successes of the theory of learning in games (with tools like regret minimization being directly used to construct superhuman-level algorithms for games like poker). This paper proposes a class of novel algorithms for extensive form game solving and both theoretically and empirically analyzes their performance (showing that this class of algorithms can have theoretical convergence rates on par with the best known algorithms). This analysis is relatively thorough and the paper is well-written and easy to read.

**Weaknesses:**

Overall, I am a little unimpressed by the results of this paper. While it is true to the best of my knowledge that this class of algorithms (as applied to extensive-form game solving) is novel, it doesn’t really seem like they unlock any new guarantees that were not previously achievable. Several times throughout the paper the authors emphasize that this is the first algorithm “based on Blackwell approachability” to achieve these guarantees, and that this resolves an interesting open question. But it is not clear to me that “based on Blackwell approachability” is really a well-defined concept (perhaps you could recover some existing algorithms via other applications of approachability) or even a desired one.

I also feel that the use of Blackwell approachability here is a little superfluous. The authors use Blackwell approachability to reduce an OLO problem on T to an OLO problem on cone(T). The eventual reduction is very simple (it essentially boils down to “projectvizing” T by adding an extra coordinate) and it is easy to see directly that the regret of the cone(T) OLO algorithm bounds the regret of the overall algorithm (that said, it is a nice observation that OLO algorithms for cone(T) seem give rise to "stepsize invariant" algorithms for original problem). It should also be pointed out that there has been significant work on Blackwell approachability since the work of Abernethy et al., including several papers which resolve some of the deficiencies the authors point out in Appendix C (which mostly stem from the fact that Abernethy et al. only consider the L_2 norm). I would recommend the authors look at “Refined approachability algorithms and application to regret minimization with global costs” by Kwon or “Pseudonorm Approachability and Applications to Regret Minimization” by Dann et al.

This would perhaps be okay if these new algorithms were shown to empirically significantly outperform state-of-the-art on some class of games, but this does not seem to be the case; if anything, they seem to underperform existing algorithms such as PCFR+. I think some of the resulting conjectures about the role of stepsize invariance (and different types of step-size of invariance) on the practical performance of these algorithms are interesting, but they are not very convincingly explored in this work.

**Questions:**

Feel free to reply to any part of the review above.

**Limitations:**

Limitations adequately addressed.

---

> ### Author Rebuttal · Authors · 2024-08-01
>
> We thank you for your time reviewing the paper. We quote your comments in italic and we respond to them in regular font.
>
> ## Response to your questions.
>
> * *But it is not clear to me that “based on Blackwell approachability” is really a well-defined concept (perhaps you could recover some existing algorithms via other applications of approachability) or even a desired one.*
>
> The algorithms derived from Algorithm 1 in our paper are derived from Blackwell approachability instances where the decision set is the treeplex $\mathcal{T}$ and the target set is the polar of the cone $\mathcal{C} = cone(\mathcal{T})$. The vector payoff is $f(x,\ell)$, given a decision $x \in \mathcal{T}$ and an instantaneous loss $\ell$. We will make this more explicit in our revised paper.
>
> At a higher level, our goal in searching for an algorithm based on Blackwell approachability was to develop algorithms that share similar principles as the RM, RM+ and PRM+ algorithms for regret minimization on the simplex, in particular, their strong stepsize invariance properties. We believe that we do achieve this goal, and that it is a desirable concept; we think it is fair to say that our algorithm is the direct analogue of the RM+ algorithm extended to the treeplex. Perhaps we should better explain that this is really what we are looking for, as opposed to *any* algorithm that could be constructed as being based on Blackwell approachability. You are right that, perhaps, one could derive other existing methods such as e.g. projected OGD with a sufficiently clever Blackwell reduction, and this would not yield the type of algorithm that we were looking for. We will better clarify this in the revised paper.
>
> * *It should also be pointed out that there has been significant work on Blackwell approachability since the work of Abernethy et al., including several papers which resolve some of the deficiencies the authors point out in Appendix C (which mostly stem from the fact that Abernethy et al. only consider the L_2 norm). I would recommend the authors look at [1] “Refined approachability algorithms and application to regret minimization with global costs” by Kwon or [2] “Pseudonorm Approachability and Applications to Regret Minimization” by Dann et al.*
>
> Thanks for pointing us to these references. We agree that [1,2] provide interesting extensions to the seminal work from [0] Abernethy et al., focusing on the case of approachability based on other (pseudo)norms. We would like to emphasize that our problem setting is different, and therefore our objectives too – we focus specifically on game solving, on fast convergence rates to Nash equilibrium, and on the stepsize properties of our algorithms. This is a fundamental difference with previous works like [1,2], which improve the reduction from [0] in other directions than ours (using other norms, focusing on regret minimization). We recognize that it is currently somewhat unknown whether game solving could benefit from Blackwell approachability based on other norms than the $\ell_2$-norm, and we will list it in our discussion section. That said, we are somewhat skeptical that alternative norms or Bregman divergences would lead to a numerical improvement. To give an analogy that we believe is fitting: we already know that the "right" distance generating function for the simplex is an entropy-based measure, or the dilated variant for EFGs, at least if we care about the ergodic rate of convergence. Yet such approaches have not performed very well numerically compared to the "Blackwell approachability-based" algorithms developed via RM+ or PRM+ run on each simplex via CFR. This is the sense in which we believe it was important to understand the performance of this type of "Blackwell approachability-based" algorithm directly on the treeplex. We also believe that some variants of the potential-based Blackwell generalizations have likely been tried, since they are known in the game-solving community. Yet, nobody ever wrote about such experiments in papers, most likely because the numerical performance was disappointing.
>
> * *This would perhaps be okay if these new algorithms were shown to empirically significantly outperform state-of-the-art on some class of games, but this does not seem to be the case; if anything, they seem to underperform existing algorithms such as PCFR+. I think some of the resulting conjectures about the role of stepsize invariance (and different types of step-size of invariance) on the practical performance of these algorithms are interesting, but they are not very convincingly explored in this work.*
>
> We agree that it would be more attractive if our algorithms outperformed the state-of-the-art. At the same time, for the reasons stated above, we do believe that it was a hole in the literature that nobody understood whether it was possible to get useful algorithms by performing some form of Blackwell approachability-style algorithm directly on the treeplex. Our results show that, indeed, such algorithms can be designed, implemented efficiently, and we explore their numerical performance. We believe that understanding the performance of such an approach was needed in the literature, although it is disappointing that we do not recover performance similar to what can be achieved via PCFR+. At the same time, given this conclusion, we also believe that identifying the new stepsize invariance conjecture is a valuable contribution toward understanding the performance of PCFR+.
>
>
> ## Conclusion.
>
> We thank you again for giving us the opportunity to improve our work. Please let us know if there are any other questions that we should address.

---

> ### Author Response · Authors · 2024-08-11
>
> As the end of the discussion period approaches, we would like to ask if our responses address your concerns and comments. We remain available to provide further clarifications on our work.

---

> > ### Comment · Reviewer_n7s4 · 2024-08-12
> >
> > Thank you for the detailed response. After reading through it (and the other reviews and comments), I've decided to maintain my current evaluation of the paper.

---

### Official Review · Reviewer_WD2M · 2024-07-15

**Soundness:** 3
**Presentation:** 3
**Contribution:** 3
**Rating:** 7
**Confidence:** 2

**Summary:**

This paper studies computations via regret minimization of Nash equilibria in zero-sum extensive form games (EFG) with the perfect recall assumption.
The actions of the players are a sequence of polytopes (treeplexes).

In prior work (counterfactual regret minimization framework), this was solved with methods that run regret minimization locally for a phase in the game on the corresponding information sets.
Common regret minimization algorithms used are regret matching that operates on the simplex and is based on Blackwell approachability, or online mirror descent (OMD).
Using regret matching as a local optimizer showed a better empirical performance.
This paper to develop Blackwell approachability-based algorithms directly for the treeplexes (instead of locally). The benefit of using Blackwell approachability is a property named "stepsize invariant", which means that stepsizes across different information sets, and the iterates of the algorithm main do not depend on them. This is different from running OMD on the treeplex.

Several algorithm instantiations are also provided and tested via numerical experiments.
The main message of the papers is to claim that information sets stepsize invariance seems like a crucial property for good empirical performance  and shed light on the strong empirical performance with such property, such as CFR+ (Counterfactual regret minimization with regret matching+ as a local optimizer)

**Strengths:**

The computation of Nash equilibria in zero-sum extensive form games via iterative methods is fundamental in "learning with games"/"self-play".
This paper studies natural approaches to tackle this problem and perhaps sheds light on an interesting property for strong empirical performance.

Although I'm not an expert in the field, the authors give enough background to explain how their approach is related to prior work.

In my understanding, this paper makes a significant contribution.

**Weaknesses:**

Couldn't find Weaknesses

**Questions:**

-

**Limitations:**

The limitations are properly addressed.

---

> ### Author Rebuttal · Authors · 2024-08-01
>
> We thank you for your time reviewing the paper and for your positive review. If you have any questions about our work, we remain available during the rebuttal period.

---

### Decision · Program_Chairs · 2024-09-25

**Decision:**

Accept (spotlight)

**Comment:**

The paper makes an interesting conceptual contribution to the topic of extensive form games (EFGs), which is a foundational and well studied problem in game-theory as well as in ML. The paper shows a somewhat surprising result regarding Blackwell approachability based algorithms and their invariance to the choice of the stepsize, which is an intriguing property that does not apply to existing algorithms towards EFGs.
From these reasons I recommend to accept this paper.